# Clinical and analytical validation of FoundationOne®CDx, a comprehensive genomic profiling assay for solid tumors

**Coren A. Milbury**[1☯], **James Creeden**[2☯*], **Wai-Ki Yip**[1], **David L. Smith**[3], **Varun Pattani**[1], **Kristi Maxwell**[4], **Bethany Sawchyn**[5], **Ole Gjoerup**[5], **Wei Meng**[1], **Joel Skoletsky**[1], **Alvin D. Concepcion**[1], **Yanhua Tang**[1], **Xiaobo Bai**[1], **Ninad Dewal**[1], **Pei Ma**[1], **Shannon T. Bailey**[1], **James Thornton**[1], **Dean C. Pavlick**[6], **Garrett M. Frampton**[6], **Daniel Lieber**[7], **Jared White**[7], **Christine Burns**[1], **Christine Vietz**[1]

1 Department Product Development, Cambridge, MA, United States of America, 2 Global Medical Affairs, Basel, MA, United States of America, 3 Department of Franchise Development, Cambridge, MA, United States of America, 4 Department of Health Economic and Outcomes Research & Payer Policy, Reimbursement, Cambridge, MA, United States of America, 5 Department of Scientific and Medical Publications, Clinical Operations, Cambridge, MA, United States of America, 6 Department of Cancer Genomics, Cambridge, MA, United States of America, 7 Department of Computational Biology, Cambridge, MA, United States of America

☯ These authors contributed equally to this work.
* jcreeden@foundationmedicine.com

**Data Availability Statement:** All relevant data were provided as supplementary information with this revised manuscript. Due to HIPAA requirements, we are not consented to share individualized

## Abstract

FoundationOne®CDx (F1CDx) is a United States (US) Food and Drug Administration (FDA)-approved companion diagnostic test to identify patients who may benefit from treatment in accordance with the approved therapeutic product labeling for 28 drug therapies. F1CDx utilizes next-generation sequencing (NGS)-based comprehensive genomic profiling (CGP) technology to examine 324 cancer genes in solid tumors. F1CDx reports known and likely pathogenic short variants (SVs), copy number alterations (CNAs), and select rearrangements, as well as complex biomarkers including tumor mutational burden (TMB) and microsatellite instability (MSI), in addition to genomic loss of heterozygosity (gLOH) in ovarian cancer. CGP services can reduce the complexity of biomarker testing, enabling precision medicine to improve treatment decision-making and outcomes for cancer patients, but only if test results are reliable, accurate, and validated clinically and analytically to the highest standard available. The analyses presented herein demonstrate the extensive analytical and clinical validation supporting the F1CDx initial and subsequent FDA approvals to ensure high sensitivity, specificity, and reliability of the data reported. The analytical validation included several in-depth evaluations of F1CDx assay performance including limit of detection (LoD), limit of blank (LoB), precision, and orthogonal concordance for SVs (including base substitutions [SUBs] and insertions/deletions [INDELs]), CNAs (including amplifications and homozygous deletions), genomic rearrangements, and select complex biomarkers. The assay validation of >30,000 test results comprises a considerable and increasing body of evidence that supports the clinical utility of F1CDx to match patients with solid tumors to targeted therapies or immunotherapies based on their tumor's genomic alterations and biomarkers. F1CDx meets the clinical needs of providers and patients to receive

patient genomic data, which contains potentially identifying or sensitive patient information and cannot be reported in a public data repository. FMI is committed to collaborative data analysis and we have well-established, and widely utilized mechanisms by which investigators can query our core genomic database of >400,000 de-identified sequenced cancers. Detailed data may be obtained by contacting the corresponding author or the Foundation Medicine Data Governance Council at data.governance.council@foundationmedicine.com.

**Funding:** This research was funded by Foundation Medicine, Inc. The funder, Foundation Medicine, Inc. (a wholly owned subsidiary of Roche) provided support in the form of salaries for all authors (CAM, JC, WKY, DLS, VP, KM, BS, OG, WM, JS, ADC, YT, XB, ND, PM, STB, JT, DCP, GMF, DL, JW, CB, CV). The funders did not have any additional role in the study design, data collection and analysis, decision to publish, or preparation of the manuscript. The specific roles of these authors were provided in the cover letter at first submission. No grants supported this study.

**Competing interests:** The authors have the following interests. At the time of this research, all authors (CAM, JC, WKY, DLS, VP, KM, BS, OG, WM, JS, ADC, YT, XB, ND, PM, STB, JT, DCP, GMF, DL, JW, CB, CV) were employed by Foundation Medicine, Inc. (a wholly owned subsidiary of Roche), the funder of this study. This does not alter the authors' adherence to all the PLOS ONE policies on sharing data and materials, as detailed online in the guide for authors.

guideline-based biomarker testing, helping them keep pace with a rapidly evolving field of medicine.

## 1. Introduction

The advances in our understanding of cancer biology over the last two decades have underscored the genomic heterogeneity among tumors of the same type, as well as similar driver mechanisms in different cancer types, enabling both tailored approaches to individual patient care as well as pan-tumor therapeutic indications [1–4]. The rapidly expanding repertoire of targeted cancer therapies provides significant challenges for oncologists treating patients, as keeping up with companion diagnostic approvals, clinical trials, and evolving understanding of cancer biology can exceed the capacity of many physicians. Testing for genomic alterations is essential for selecting the most appropriate targeted treatments or immunotherapies and clinical trials, and for avoiding futile treatments where resistant tumors are present or actionable targets are absent [5, 6].

The ability to treat cancer patients with precision medicine requires an approach to identify targetable alterations which may occur in many different genes, alteration types and complex genomic signatures. With the proliferation of highly effective targeted therapies, determining the best treatment may not always be possible by examining the patient or running a single-analyte test, but rather requires a molecular diagnostic that is both time- and tissue-efficient while also delivering accurate and reproducible performance. Highly validated testing solutions that comprehensively cover actionable alterations and deliver the latest literature-based guidance on potential treatment options, along with relevant clinical trials and levels of evidence, offer a streamlined and comprehensive solution to oncologists and provide patients with reassurance that even rare but potentially treatable alterations can be identified [7–13]. Hot-spot and smaller-panel testing approaches require a-priori assumptions about the alterations that may be present and can miss actionable alterations that are beyond their narrow scope [14], including novel alterations that may be treatable or confer resistance, such as oncogene fusions [15–20]. In contrast, broader whole exome sequencing (WES; note all abbreviations/acronyms are available in the S1 Appendix, Table S1 in S1 Appendix [all supplemental tables are located in S1 Appendix]) or whole genome sequencing (WGS) raise different challenges for clinicians. WES evaluates all exonic protein-coding regions of the human genome, encompassing ~30 million base pairs (~1% of the human genome), while WGS aims to sequence the entire genome. These efforts produce data from an excessive breadth of the genome, but generally deliver lower depth of sequencing that results in a higher LoD relative to broad-panel assays [21, 22]. Further, WES and WGS approaches can entail higher cost, complex analytics, and longer turnaround time, but critically, they may also burden the report recipient with many variants of unknown significance (VUS) from genes not relevant to therapeutic decision-making for the patient [23–27].

CGP approaches are an increasingly valuable and important part of the molecular characterization of the tumor and subsequent selection of the most relevant treatment options for patients with advanced cancer, and can bridge the gap between the narrow focus of single-marker or hot-spot tests and the lower sensitivity of WES or WGS. Clinical practice guidelines support use of CGP for an increasing number of cancer types including non-small cell lung cancer (NSCLC) [28], melanoma [29], breast [30], colorectal cancer (CRC) [31, 32], endometrial uterine [33], and gastroesophageal cancer [34]. A CGP approach utilizes massively parallel

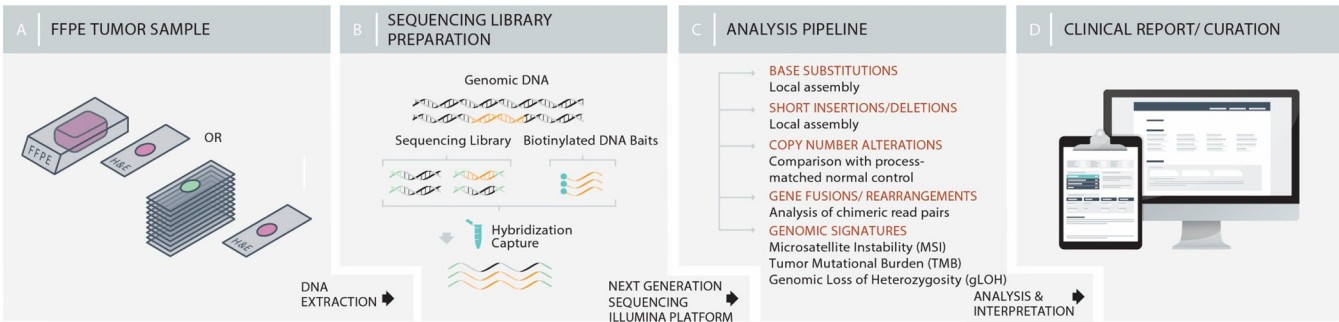

**Fig 1. FoundationOne®CDx workflow.** CAP = College of American Pathologists; CDx = companion diagnostic; CLIA = Clinical Laboratory Improvement Amendments; DNA = deoxyribonucleic acid; FDA = United States Food and Drug Administration; FFPE = formalin-fixed, paraffin-embedded.

NGS to ensure detection of all alteration types, including novel alterations, in a broad but highly relevant panel of genes, to allow greater depth of sequencing and sensitivity while minimizing potentially confusing and extraneous information from irrelevant genes on the clinical report. Foundation Medicine, Inc (Foundation Medicine)'s CGP approach for formalin-fixed, paraffin-embedded (FFPE)-based solid tumor testing, F1CDx, utilizes highly sensitive and validated hybrid capture-based NGS technology to examine entire regions of 324 cancer-relevant genes for all solid tumor types (Fig 1). Unlike smaller, disease-specific panel tests or more focused "hot-spot" tests, this comprehensive approach enables identification of all four main classes of genomic alterations: SUBs, INDELs, CNAs, and gene fusions/rearrangements along with increasingly complex genomic signatures, including MSI, TMB, and gLOH. F1CDx achieves this with high sensitivity in a single assay minimizing the total amount of tissue required while maximizing informative testing [7–13].

While NGS testing is broadly available for research purposes, the application of this technology to clinical decision-making by ensuring robustness, accuracy, reproducibility, and utility is technically and economically challenging. Providers of NGS testing services for cancer have multiplied, but these services are widely undervalued [35], and robust evidence of clinical validity and utility are scarce. The National Institutes of Health lists over 77,364 genetic tests for 17,285 conditions across 18,731 genes at 549 labs, while the US FDA lists only 25 non-imaging-based companion diagnostics [36, 37]. Of these 25, only 7 offer results for more than one gene, and three are from Foundation Medicine. This disparity in breadth of gene coverage undermines both public confidence and reimbursement of high-complexity testing. With an abundance of choice, and absence of standardized quality measures and published validation and utility data, payers may be inclined to substitute cheaper, less robust tests, and patients may be deprived of the opportunity to benefit from potentially life-extending therapies as a result of false-negative results, and/or be exposed to unnecessary adverse events and costs through false-positive results [15, 38–41]. Extensive validation and regulatory review are

critical safeguards to protect patients from potentially harmful tests and wasted resources [42]. It is crucial to apply rigorous standards consistently across tests with high-risk intended uses such as those used for cancer management. The rigor of the FDA's premarket review—a process grounded in the highest standards of scientific and medical integrity and transparency—builds trust. By enforcing high evidentiary standards for these high-risk tests, regulatory authorities minimize the risks to patients and payers while enabling access to the most reliable test providers [43], as demonstrated by the FDA/Centers for Medicare & Medicaid Services (CMS) parallel review process [44].

The FDA's Center for Drug Evaluation and Research plays a crucial role in assuring access to safe and effective medicines based on a minimal standard of evidence that demonstrates a drug is likely to be beneficial, and not excessively harmful, to a given treatment population [45]. As treatment costs rise and criteria for selecting eligible treatment populations become more complex, the quality of methods for testing those criteria is under increased scrutiny. For complex predictive biomarkers like MSI, TMB, and gLOH, high quality standards for defining pre-analytics, sample handling, and validation are especially important, as a growing number of therapeutic decisions are influenced by these signatures [24, 46, 47]. Further, for complex biomarkers, such as TMB, it is essential to use a broad-panel test that comprehensively evaluates a large amount of genomic content and that is well-validated across tumor types. FDA approvals of companion diagnostics in a pan-tumor setting for TMB ≥10 mutations per megabase (mut/Mb) for pembrolizumab and *NTRK*-fusions for larotrectinib demonstrate the requirement for CGP in order to optimize therapeutics in oncology and ensure patients have the opportunity to access to potentially beneficial therapy [28–34, 48–53].

The evidence used to demonstrate robustness of a diagnostic includes analytical validity, clinical validity, and clinical utility. Analytical validity of an NGS assay refers to how accurate and reliable the test is in identifying a particular analyte, such as a genomic alteration or genomic signature [54]. Clinical validity refers to a companion diagnostic's capability to predict, measure, or detect a clinical condition or status for its intended use [55]. While clinical validity refers to the relationship between an analyte and the presence or absence of a particular disease state, clinical utility refers to the ability of test results to predict improved health outcomes from an intervention [54]. The FDA requires robust evidence of analytical and clinical validation, exceeding standards set by Clinical Laboratory Improvement Amendments (CLIA), prior to approval or clearance of a diagnostic device, in order to assure consumers of the quality of results and ensure tests can be used to provide information to support the diagnosis, treatment, management, or prevention of a disease [56]. The FDA approvals of validated clinical trial assays (CTAs) as companion diagnostics support improved therapeutic results and directly demonstrate both clinical validity and clinical utility. The FDA also enables test providers to develop companion diagnostic claims through clinical bridging when trial specimens are available, or through non-inferiority studies when clinical trial specimens are not available. Through this pathway, companion diagnostic claims can be established for validated assays that have been demonstrated to be concordant or non-inferior to the clinical trial assays that demonstrated utility in pivotal trials. Further, a candidate companion diagnostic can demonstrate the ability to predict clinical efficacy of a corresponding therapeutic with comparable efficacy to that observed in the associated clinical trial. For example, given an observed therapeutic efficacy, if the candidate companion diagnostic were used to identify patients that would respond to treatment, instead of the clinical trial assay, the observed therapeutic efficacy will have been the same.

FDA's Center for Devices and Radiological Health [57] is responsible for establishing and ensuring quality standards by evaluating and regulating the performance of the companion diagnostics [58], particularly for high-risk testing services such as those for cancer genotyping

for therapeutic decision-making [59]. Foundation Medicine profiles >100,000 tumor specimens each year across its laboratories in Cambridge, MA, USA, Research Triangle Park, NC, USA, and Penzberg, Germany. As Foundation Medicine is both the manufacturer and laboratory service provider, rigorous validation is performed to meet the requirements of both the FDA and CLIA.

The analyses presented here describe the broad analytical and clinical validation of F1CDx (Foundation Medicine; Cambridge, MA), a leading CGP platform. The validations of MSI, TMB, and gLOH have been further described in detail elsewhere [60–62].

## 2. Methods and results

### FoundationOne®CDx assay

F1CDx is an FDA-approved NGS-based *in vitro* diagnostic device that targets 324 cancer-related genes. F1CDx is performed exclusively by Foundation Medicine as a central laboratory testing service using DNA extracted from FFPE tumor samples from cancer patients [63]. The laboratory components are performed within Foundation Medicine's College of American Pathologists (CAP)-accredited laboratory that has been certified by CMS pursuant to CMS procedures for the certification of international laboratories in accordance with CLIA. The F1CDx assay is regulated by the FDA in accordance with FDA's standard of a reasonable assurance of safety and effectiveness, together with FDA's Quality System regulations (21 CFR Part 820). Additional clinical decision insights and genomic analyses are provided as a professional laboratory service under CLIA and CAP regulations.

F1CDx is the result of the evolution of Foundation Medicine's original FoundationOne® assay [64]. FoundationOne® testing has been used for clinical testing since 2012. The assay workflow is presented in Fig 1. The assay employs a single DNA extraction method from routine FFPE biopsy or surgical resection specimens; 50–1,000 ng of DNA undergo whole-genome shotgun library construction and hybridization-based capture of all coding exons from 309 cancer-related genes, including one promoter region, one non-coding (ncRNA), and select intronic regions from 34 commonly rearranged genes, 21 of which also include the coding exons (refer to Tables S2 and S3 in S1 Appendix for the complete list of genes included in F1CDx). Using an Illumina® NGS platform, hybrid capture-selected libraries are sequenced to high uniform depth (targeting >500X median coverage with >99% of exons at coverage >100X). Sequence data are then processed using a purpose-built analysis pipeline designed to detect all classes of genomic alterations, including SVs, CNAs (amplifications and homozygous gene deletions), and select genomic rearrangements (e.g., gene fusions). Genomic signatures including MSI and TMB are reported, in addition to positive homologous recombination deficiency (HRD) status (tumor *BRCA*-positive and/or gLOH high) in ovarian cancer.

### Bioinformatic methods

Sequence data are analyzed using a proprietary software system developed by Foundation Medicine. Sequence data are mapped to the human genome (hg19) using Burrows-Wheeler Aligner (BWA; a program for aligning sequence reads with large-scale reference genomes) [65]. Polymerase chain reaction (PCR) duplicate read removal and sequence metric collection are performed using Picard and SAMtools [66]. Variant calling is performed only in genomic regions targeted by the test.

For the detection of SVs and rearrangements, a *de novo* assembly is performed. This is done using software to generate a de Bruijn graph including all k-mers in reads mapping to a particular locus [67]. For each variant, there is a set of k-mers supporting the variant and a set of k-mers that would support the reference or another variant at the location. Each candidate

variant is then scanned against reads in the locus to identify which support the candidate variant, a different variant, or the reference at the location. The supporting reads for each candidate variant are analyzed, and metrics used to evaluate the quality of the variant call are calculated. The final variant calls are made based on a series of quality control (QC) filters, which will reject a call based on the intrinsic sample noise, the expected noise level for the particular variant, and other known error modes (e.g., sequence homology).

Short variants are reported as a frequency relative to overall depth of coverage; this is denoted as mutation allele frequency (MAF). Alterations are classified as "known," "likely," or "unknown" based on their status in COSMIC. Alterations classified as "other" include truncating events in tumor suppressor genes (splice, frameshift, and nonsense) as well as variants that appear in hot-spot locations but do not have a specific COSMIC association. Variants may be classified as VUS when the significance and impact upon cancer progression is unknown due to a lack of reported evidence and conclusive change in function. It is recognized that some genetic alterations and variants will not impact functionality and do not increase cancer risk.

Genomic rearrangements are identified by analyzing chimeric read pairs. Chimeric read pairs are defined as read pairs for which reads map to separate chromosomes, or at a distance of over 2 kilobase pairs (kbp). Pairs are clustered by genomic coordinate of the pairs, and clusters containing at least five chimeric pairs (three for known fusions) are identified as rearrangement candidates. Filtering of candidates is performed by mapping quality (MQ >30) and distribution of alignment positions (standard deviation >10). Rearrangements are annotated for predicted function (e.g., creation of fusion gene).

CNAs are detected using a comparative genomic hybridization-like method [68]. First, a log-ratio profile of the sample is obtained by normalizing the sequence coverage obtained at all exons and genome-wide single-nucleotide polymorphisms (SNPs) against a process-matched normal control. This profile is segmented and interpreted using allele frequencies of sequenced SNPs to estimate tumor purity and copy number at each segment.

Tumor content and purity of a sample are derived separately. Board-certified pathologists will assess tumor content through the enumeration of nucleated tumor cells. The assay requires greater than 20% nucleated tumor cells to enter into the DNA extraction procedure. This upstream assessment is complemented by the downstream computational calculation of tumor purity just described above. The computational tumor purity assessment is calculated based off SNP allele frequencies and is also used to inform the accuracy of copy number modeling and the calling for several complex biomarkers. The underlying copy number modeling used in making these estimations is a simple 2-component system consisting of a mixture of normal diploid cells and aneuploid tumor cells, where the tumor purity corresponds to the fraction of tumor cells in the mixture. The aneuploidy of the tumor cells is modeled as an integral copy number level for each allele of each segment.

To determine MSI status, repetitive loci (minimum of five repeat units of mono-, di-, and trinucleotides) are assessed to determine what repeat lengths are present in the sample. A locus containing a repeat length not present in an internal database generated using >3,000 clinical samples is considered to be unstable. An MSI indicator is generated by calculating the fraction of unstable loci, considering only those loci that achieve adequate coverage for consideration for the sample.

TMB is measured by counting coding SVs present at ≥5% allele frequency and filtering out potential germline variants according to published databases of known germline polymorphisms including Single Nucleotide Polymorphism Database (dbSNP) and Genome Aggregation Database (gnomAD). Additional germline alterations are assessed for potential germline status and filtered out using a somatic-germline/zygosity (SGZ) algorithm [69]. Furthermore, known and likely driver mutations are filtered out to exclude bias. The resulting mutation

number is then divided by the coding region corresponding to the number of total variants counted, or approximately 790 kilobases (kb); the resulting number is reported in units of mut/Mb. The clinical validity of TMB defined by this panel has been established for TMB-high (TMB-H) as ≥10 mut/Mb as a qualitative status.

For ovarian tumor samples, gLOH is measured by the percentage of loss of heterozygosity (LOH) in the tumor genome. To compute gLOH for each tumor, LOH regions are inferred across the 22 autosomal chromosomes using the genome-wide copy number profile and minor allele frequencies of the germline SNPs. Certain LOH regions are excluded from analysis, including: 1) LOH regions spanning ≥90% of a whole chromosome or chromosome arm, as these LOH events are likely due to non-HRD mechanisms; and 2) regions in which LOH inference is ambiguous. For each tumor, the percentage of the genome with LOH is computed as 100 times the total length of non-excluded LOH regions divided by the total length of non-excluded regions of the genome. gLOH ≥16 is defined as "LOH high," gLOH <16 is "LOH low," and an indeterminable result is "LOH unknown." In some cases, due to QC issues such as low tumor purity, noisy CNA data, and contamination that may affect copy number modeling, it is not possible to accurately calculate LOH. In such cases, LOH will be reported as "unknown."

## Sample selection

In total, >30,000 test results have been evaluated in F1CDx assay validation studies. Samples used for analytical and clinical validation studies consisted of FFPE solid tumor specimens from patients with cancer, as well as DNA samples selected from an inventory of residual banked DNA isolated from FFPE tumor specimens, including >56 tumor tissue types. Common and rare alterations, including SUBs, INDELs, rearrangements, and CNAs, have been evaluated across the genomic breadth of the baitset, covering >300 genes. The extensive pan-tumor validation supports the clinical utility of the F1CDx test, which evaluates over 500 cancer disease ontologies. Institutional Review Board (IRB) approval was obtained from the New England IRB prior to use of samples in the described validation studies.

In some instances, such as the evaluation of CNAs or complex biomarkers, targeted allele frequencies were achieved by titrating DNA samples into fragmented genomic DNA derived from peripheral blood mononuclear cells.

## 3. Clinical and analytical validity of F1CDx

As presented in Table 1, F1CDx is an FDA-approved companion diagnostic across two pan-tumor indications (*NTRK* gene fusions and TMB-high [TMB-H]) as well as in NSCLC, breast,

**Table 1. F1CDx companion diagnostic claims.**

| Tumor type | Biomarker(s) detected | Therapy |
|---|---|---|
| Pan-tumor | TMB ≥10 mutations per megabase | Keytruda® (pembrolizumab) |
| | *NTRK1/2/3* fusions | Vitrakvi® (larotrectinib) |
| NSCLC | *EGFR* exon 19 deletions and *EGFR* exon 21 L858R alterations | Gilotrif® (afatinib), Iressa® (gefitinib), Tagrisso® (osimertinib), or Tarceva® (erlotinib) |
| | *EGFR* exon 20 T790M alterations | Tagrisso® (osimertinib) |
| | *ALK* rearrangements | Alecensa® (alectinib), Alunbrig® (brigatinib), Xalkori® (crizotinib), or Zykadia® (ceritinib) |
| | *BRAF* V600E | Tafinlar® (dabrafenib) in combination with Mekinist® (trametinib) |
| | *MET* exon 14 skipping | Tabrecta™ (capmatinib) |

*(Continued)*

**Table 1.** (Continued)

| Tumor type | Biomarker(s) detected | Therapy |
|---|---|---|
| Breast cancer | *ERBB2* (HER2) amplification | Herceptin® (trastuzumab), Kadcyla® (ado-trastuzumab-emtansine), or Perjeta® (pertuzumab) |
| | *PIK3CA* C420R, E542K, E545A, E545D [1635G>T only], E545G, E545K, Q546E, Q546R, H1047L, H1047R, and H1047Y alterations | Piqray® (alpelisib) |
| CRC | *KRAS* wild-type (absence of detectable mutations in codons 12 and 13) | Erbitux® (cetuximab) |
| | *KRAS* wild-type (absence of detectable mutations in exons 2, 3, and 4) and *NRAS* wild type (absence of detectable mutations in exons 2, 3, and 4) | Vectibix® (panitumumab) |
| Ovarian cancer | *BRCA1/2* alterations | Lynparza® (olaparib) or Rubraca® (rucaparib) |
| Prostate cancer | HRR gene (*BRCA1*, *BRCA2*, *ATM*, *BARD1*, *BRIP1*, *CDK12*, *CHEK1*, *CHEK2*, *FANCL*, *PALB2*, *RAD51B*, *RAD51C*, *RAD51D*, and *RAD54L*) alterations | Lynparza® (olaparib) |
| Melanoma | *BRAF* V600E | Tafinlar® (dabrafenib) or |
| | | Zelboraf® (vemurafenib) |
| | *BRAF* V600E and V600K | Mekinist® (trametinib) or Cotellic® (cobimetinib) in combination with Zelboraf® (vemurafenib) |
| Cholangiocarcinoma | *FGFR2* fusions and select rearrangements | Pemazyre™ (pemigatinib) |
| | | Truseltiq™ (infigratinib) |

CRC = colorectal cancer; HRR = homologous recombination repair; NSCLC = non-small cell lung cancer; TMB = tumor mutational burden.

CRC, ovarian, prostate, melanoma, and cholangiocarcinoma. Importantly, as the companion diagnostic indications for F1CDx are constantly evolving, the FDA's list of cleared or approved companion diagnostic devices provides a complete list of the currently approved FDA companion diagnostic claims, including all approved companion diagnostic claims for F1CDx [70]. Each of the companion diagnostic claims for F1CDx were FDA-approved based upon the clinical validity as determined by use of F1CDx as the clinical trial assay, non-inferiority concordance testing against FDA-approved diagnostics for that indication, retrospective or prospective analyses, or clinical bridging studies.

## Clinical validation: Non-inferiority

F1CDx was clinically validated through the performance of either clinical bridging or non-inferiority. Using non-inferiority, the clinical validity of the F1CDx assay as a companion diagnostic was evaluated for identifying patients with specific disease indications to evaluate patient eligibility for treatment by targeted therapies for defined biomarkers. Concordance between F1CDx and a validated orthogonal comparator assay was assessed through the evaluation of negative percent agreement (NPA) and positive percent agreement (PPA). The orthogonal concordance assay was considered the gold standard within each non-inferiority analysis. Samples were tested across two replicates (denoted as CCD1 and CCD2) by the comparator test and one replicate by F1CDx, as described by Li et al [76]. Detailed concordance results and sample numbers are provided in supplementary tables. The clinical validation performed via the demonstration of non-inferiority is summarized in Table 2.

## Clinical validation: Clinical bridging

In addition to non-inferiority, the F1CDx assay was clinically validated through clinical bridging analyses to establish clinical utility. In each scenario, the clinical efficacy as demonstrated by the local clinical trial assays (CTAs) was compared to the clinical efficacy as demonstrated by the F1CDx assay. The clinical bridging validation is summarized in Table 3. In addition, clinical utility of the F1CDx assay was also demonstrated in a pan-tumor setting for TMB-H

**Table 2. Clinical validity of F1CDx via non-inferiority for companion diagnostic claims.**

| Disease Indication | Biomarker | Therapy | PPA, % (95% CI) | NPA, % (95% CI) | Comparator Assay | Detailed Concordance Results |
|---|---|---|---|---|---|---|
| NSCLC | *EGFR* exon 19 deletions and exon 21 L858R | afatinib, gefitinib, osimertinib, erlotinib | 98.1 (93.5, 99.8) | 99.4 (96.4, 100.0) | cobas® EGFR Mutation Test v2 (Roche Molecular Systems) | Tables S4-S6 in S1 Appendix |
| NSCLC | *EGFR* T790M | osimertinib | 98.9 (93.8, 100.0) | 86.1 (78.1, 92.0) | cobas® *EGFR* mutation Test v2 (Roche Molecular Systems) | Tables S7-S9 in S1 Appendix |
| NSCLC | *ALK* rearrangements[a] | alectinib, crizotinib, ceritinib, brigatinib | 92.9 (85.1, 97.3) | 100 (95.2, 100.0) | Ventana ALK (D5F3) CDx Assay | Tables S10-S12 in S1 Appendix |
| | | | | | Vysis *ALK* Break Apart FISH Probe Kit | |
| Breast cancer | *ERBB2* (HER2) amplification | trastuzumab, ado-trastuzumab emtansine, pertuzumab | 89.4 (82.2, 94.4) | 98.4 (95.3, 99.7) | HER2 FISH PharmDx® Kit (Dako Denmark, A/S) | Tables S13-S15 in S1 Appendix |
| CRC | *KRAS* wild-type | cetuximab, panitumumab | 100 (97.9, 100.0) | 100 (97.6, 100.0) | therascreen® KRAS RGQ PCR Kit (QIAGEN) | Tables S16-S18 in S1 Appendix |
| Melanoma | *BRAF* V600 mutation | dabrafenib, vemurafenib, trametinib, cobimetinib in combination with vemurafenib | 99.4 (166/167) | 89.6 (121/135)[b] | cobas® 4800 BRAF V600 mutation test (Roche Molecular Systems, Inc) | Tables S19-S21 in S1 Appendix |
| | *BRAF* V600E mutation | | 99.3 (149/150) | 99.2 (121/122) | | |
| Melanoma | *BRAF* V600 dinucleotide | dabrafenib, vemurafenib, trametinib, cobimetinib in combination with vemurafenib | 96.3 (26/27) | 100 (24/24) | THxID® BRAF Kit (bioMérieux) | Table S22 in S1 Appendix |

CTA = clinical trial assay; FISH = fluorescence *in situ* hybridization; NPA = negative percent agreement; PPA = positive percent agreement; PCR = polymerase chain reaction.

[a] Samples evaluated were from a phase 3, multicenter, open-label study (NCT02075840) that evaluated the efficacy and safety of alectinib compared with crizotinib in treatment-naïve cancer patients with *ALK* rearrangements.

[b] The reported difference in NPA values for *BRAF* V600 and *BRAF* V600E is likely attributed to known sensitivity differences in the cobas *BRAF* mutation test, which has lower sensitivity for detection of dinucleotide V600 alterations than for the single nucleotide V600E c.1799T>A alteration, especially for samples in which F1CDx detected the nucleotides to be of lower than 40% mutational allele frequency, leading to low NPA values.

($\geq$10 mut/MB), *NTRK* fusions, and MSI. Clinical utility of F1CDx comprehensive genomic profiling was further demonstrated for deleterious alterations within a panel of homologous recombination repair (HRR) genes in patients with castration-resistant prostate cancer. The F1CDx assay's clinical efficacy was supported by bridging to the Foundation Focus CDx$_{BRCA}$ $_{LOH}$ assay in ovarian cancer patients with alterations in *BRCA1*, *BRCA2*, or demonstrated genomic loss of heterozygosity (gLOH). Detailed results are described below for TMB-H, HRR, and gLOH.

**Clinical validation for detection of TMB-H ($\geq$10 mut/Mb) in solid tumors.** Of the 1,050 patients with select advanced solid tumors enrolled in the KEYNOTE-158 phase 2 basket trial of pembrolizumab and included in the efficacy analysis population, 790 patients had sufficient tissue for testing with F1CDx and were included in the retrospective analysis of this clinical data [49]. The primary efficacy outcome for pembrolizumab was objective response rate (ORR), defined as the proportion of patients in the analysis population who had a response (complete response [CR] or partial response [PR]) per central review as assessed by Response Evaluation Criteria in Solid Tumors (RECIST) v1.1 [71]. For the purposes of outcomes reported hereafter, objective and overall response rate are used interchangeably and are abbreviated as ORR. TMB-H ($\geq$10 mut/Mb) was associated with a clinically meaningful improvement in patients with previously treated solid tumors, as demonstrated by an ORR (primary endpoint) of 29.4% (95% confidence interval [CI]: 20.8, 39.3) [72], compared to 6.3% (95% CI: 4.6, 8.3) in the non-TMB-H group in the therapeutic efficacy (TE) population as determined by F1CDx (Table 4). The results reported with F1CDx were similar to the clinical efficacy

outcomes as reported in the device validation (DV) population (defined in the Supplementary Methods in S2 Appendix) and recently compared to the efficacy analysis of over 1,700 patients representing over 20 tumor types with TMB determined by WES who were treated with pembrolizumab in a series of 12 clinical trials. The response to pembrolizumab was higher in the WES TMB-H population (TMB $\geq$175 mut/exome is approximately equivalent to $\geq$10 mut/ Mb by F1CDx used in KEYNOTE-158) compared to <175 mut/exome population, with an observed ORR of 31.4% (95% CI: 27.1, 36.0) by BICR (Blinded Independent Central Radiology Review) versus 9.5% in the WES TMB <175 mut/exome group [73].

## Clinical validation: Ovarian cancer

**Clinical validation for detection of *BRCA1/2* alterations in ovarian cancer.** The clinical performance of F1CDx for *BRCA1/2* classification was established based on available tumor

**Table 3. Clinical utility of F1CDx in each efficacy analysis population.**

| Therapy (References) | Clinical Trials | Biomarker | Disease Indication | Concordance to Local CTAs | | | Clinical Efficacy | | |
|---|---|---|---|---|---|---|---|---|---|
| | | | | PPA, % (95% CI) | NPA, % (95% CI) | Detailed Data | Clinical Endpoints | F1CDx Results | CTA Results |
| larotrectinib[a,b,c] ([48, 74] [75]) | LOXO-TRK-14001 (Bayer 20288, NCT02122913) LOXO-TRK-15002 (Bayer 20289, NAVIGATE, NCT02576431) LOXO-TRK—15003 (Bayer 20290, SCOUT, NCT02637687) | Gene fusions in *NTRK1*, *NTRK2*, and *NTRK3* | Solid tumors | 84.1 (69.9, 93.4) | 100.0 (98.4, 100.0) | Tables S23 and S24 in S1 Appendix | ORR, % (n/N) (95% CI) | 77 (20/ 26) (56, 91) N = 26 | 75 (41/ 55) (61, 85) N = 55 |
| | | | | | | | DOR, range (months) % with duration $\geq$6 months % with duration $\geq$9 months % with duration $\geq$12 months | 1.6, 20.3 80.0 65.0 25.0 | 1.6, 33.2 73.2 63.4 39.0 |
| capmatinib[d,e] ([77–79] [80]) | GEOMETRY-mono 1 trial [79] | *MET* SNVs and INDELs that lead to exon 14 skipping | NSCLC | 98.6[f] (92.6, 100.0) | 100.0[f] (97.1, 100.0) | Table S25 in S1 Appendix | Cohort 4: ORR[g], % (n/ N) (95% CI) | 44.2 (23/ 52) (30.5, 58.7) N = 52 | 40.6 (28/ 69) (28.9, 53.1) N = 69 |
| | | | | | | | Cohort 5b: ORR[g], % (n/ N) (95% CI) | 70.0 (14/ 20) (45.7, 88.1) N = 20 | 67.9 (19/ 28) (47.6, 84.1) N = 28 |
| | | | | | | | Cohort 4: Median DOR[g], months (95% CI) Patients with DOR >12 months, % | 9.72 (4.27, 12.98) 34.8 N = 23 | 9.7 (5.5, 13.0) 32 N = 28 |
| | | | | | | | Cohort 5b: Median DOR[g], months (95% CI) Patients with DOR >12 months, % | 12.58 (5.55, 25.33) 50.0 | 12.6 (5.5, 25.3) 47 |

*(Continued)*

**Table 3.** (Continued)

| Therapy (References) | Clinical Trials | Biomarker | Disease Indication | Concordance to Local CTAs | | | Clinical Efficacy | | |
|---|---|---|---|---|---|---|---|---|---|
| | | | | PPA, % (95% CI) | NPA, % (95% CI) | Detailed Data | Clinical Endpoints | F1CDx Results | CTA Results |
| pemigatinib [89] [90] | FIGHT-202 | *FGFR2* fusions and select rearrangements | CCA | 100.0 [h] (95.70, 100.00) | 100.0 [h] (96.27, 100.00) | Table S26 in S1 Appendix | ORR[i], % (95% CI) | 37.50 (26.92, 49.04) N = 80 | 35.51 (26.50, 45.35) N = 107 |
| infigratinib [45] | CBGJ398X2204 (NCT02150967) | *FGFR2* fusions and select rearrangements | CCA | 96.7 (88.6, 99.1) | 100.0 (96.4, 100.0) | Table S27 in S1 Appendix | ORR[i], % (95% CI) | 28.07 (17.22, 38.92) N = 67 | 23.15 (16.20, 31.94) N = 108 |
| alpelisib + fulvestrant [81, 82] | SOLAR-1 clinical trial | *PIK3CA* alterations (*PIK3CA* C420R, E542K, E545A, E545D [1635G>T only], E545G, E545K, Q546E, Q546R, H1047L, H1047R, and H1047Y) | Breast cancer | 93.8 (87.7, 97.5) [j] | 98.8 (95.6, 99.8) [j] | Table S28 in S1 Appendix | PFS[k], months, HR (95% CI) CTA1[j] | 11.2 0.52 (0.29, 0.93) N = 56 | 11.0[l] 0.65 (0.50, 0.85) N = 169 |
| | | | | 91.6 (87.1, 95.0) [j] | 98.8 (95.7, 99.9) [j] | | PFS[k], months, HR[e] (95% CI) CTA2[j] | 10.9 0.35 (0.16, 0.77) N = 42 | |

BICR = blinded independent central review; CCA = cholangiocarcinoma; CI = confidence interval; CTA = clinical trial assay; DOR = duration or response; F1CDx = FoundationOne®CDx; FISH = fluorescence *in situ* hybridization; HR = hazard ratio; NPA = negative percent agreement; NR = not reported; NSCLC = non-small cell lung cancer

PFS = Progression Free Survival; PCR = polymerase chain reaction; PPA = positive percent agreement; RECIST = Response Evaluation Criteria in Solid Tumors; RT-PCR = reverse transcriptase-polymerase chain reaction.

[a] PPA and NPA results exclude the F1CDx invalid results. Including the F1CDx invalid results, the PPA was 82.2% (95% CI: 67.9, 92.0) and the NPA was 98.3% (95% CI: 95.6, 99.5).

[b] ORR was assessed by an independent review committee using RECIST v1.1 [71].

[c] Local CTAs included DNA NGS, RNA NGS, FISH, and RT-PCR methods, with the majority (92%) of the clinical trial patients with known *NTRK* fusion status enrolled with NGS methods.

[d] Using an RT-PCR CTA, Cohort 4 enrolled 69 patients with *MET* exon 14 skipping alterations and one or two prior lines of therapy, while Cohort 5b enrolled 28 patients with *MET* exon 14 skipping alterations who were treatment naïve. F1CDx was used to analyze samples retrospectively from patients enrolled in the GEOMETRY-mono 1 trial.

[e] The results exclude the F1CDx invalid results. Including the F1CDx invalid results, the PPA was 92.3% (95% CI: 84.0, 97.1) and the NPA was 99.2% (95% CI: 95.7, 100.0).

[f] The concordance reported is for the combined cohorts (Cohort 4 and Cohort 5b).

[g] Cohort 4: previously treated patients; Cohort 5b: treatment-naïve patients. ORR as assessed by BICR according to RECIST v1.1. DOR is based on the data reported in the capmatinib USPI.

[h] The enrollment assay was the FoundationOne assay, an earlier version of the F1CDx assay.

[i] ORR per central review per RECIST v1.1. Note that ORR is objective response rate for pemigatinib and overall response rate for infigratinib.

[j] CTA1 = PCR-based PIK3CA hot-spot test; CTA2 = PCR-based *PIK3CA* hot-spot test. The results shown exclude the F1CDx invalid results. Including the F1CDx invalid results, the CTA1 PPA was 93.0% (95% CI: 86.6%, 96.9%) and the CTA1 NPA was 95.8% (95% CI: 91.5%, 98.3%). Including the F1CDx invalid results, the CTA2 PPA was 90.4% (95% CI: 85.7%, 93.9%) and the CTA2 NPA was 97.0% (95% CI: 93.2%, 99.0%).

[k] PFS by investigator assessment in patients with *PIK3CA* alteration-positive tumors. The HR shown here for both the F1CDx results and the CTA results is for alpelisib + fulvestrant for risk of disease progression or death compared to placebo in the *PIK3CA* alteration-positive population.

[l] The CTA results report the combined efficacy of both CTA1- and CTA2-enrolled patients.

analysis using FoundationFocus™ CDx_BRCA, (which is a precursor to the F1CDx device) in the ARIEL2 and Study 10 clinical studies of rucaparib and via F1CDx in the clinical study D0818C00001 (SOLO1) of olaparib [83, 84].

**Table 4. Clinical utility of F1CDx for determining TMB-H status ($\geq$10 mut/Mb) in solid tumors in the efficacy analysis population.**

| Clinical endpoint | Clinical efficacy of pembrolizumab | | | |
|---|---|---|---|---|
| | TE population (F1CDx results) TMB $\geq$10 mut/Mb | TE population (F1CDx results) TMB <10 mut/Mb | DV population TMB $\geq$10 mut/Mb | DV population TMB <10 mut/Mb |
| | n = 102 | n = 688 | n = 91 | n = 628 |
| ORR[a], % (n/N) | 29.4 (30/102) | 6.3 (43/688) | 33.0 (30/91) | 6.5 (41/628) |
| (95% CI) | (20.8, 39.3) | (4.6, 8.3) | (23.5, 43.6) | (4.7, 8.8) |

CI = confidence interval; CR = complete response; DV = device validation; F1CDx = FoundationOne®CDx; mut/Mb = mutations per megabase; ORR = objective response rate; PR = partial response; TE = therapeutic efficacy; TMB = tumor mutational burden; WES = whole exome sequencing.

[a] Central radiology assessed responses per RECIST 1.1 (confirmed).

F1CDx and FoundationFocus™ CDx$_{BRCA}$ assays are equivalent with the exception of an updated analysis pipeline in use for F1CDx and reporting software that allowed for comprehensive reporting of all relevant alterations detected by the F1CDx platform [84]. Comprehensive validation of the analysis pipeline, which included robust regression testing and reanalysis of FoundationFocus™ CDx$_{BRCA}$ clinical bridging sample data, was performed. The assays were determined to be concordant for determining HRD status for patients who benefited from rucaparib treatment (Table S29 in S1 Appendix).

ARIEL2 (NCT01891344) was a two-part, phase II, open-label study of oral rucaparib and Study 10 (NCT01482715) was a three-part, open-label, phase I/II study of oral rucaparib. The clinical validation of FoundationFocus™ CDx$_{BRCA}$ was based on clinical bridging retrospective analysis of 149 patients. ARIEL3 (NCT01968213) was a double-blind, placebo-controlled, phase 3 trial of rucaparib that enrolled 564 women with platinum-sensitive, high-grade ovarian, fallopian tube, or primary peritoneal cancer [85]. The clinical validation of the expanded version of test, FoundationFocus™ CDx$_{BRCA\ LOH}$, which introduced the assessment of gLOH, was based on clinical bridging retrospective analysis of 518 of the 564 patients. Using the expanded test FoundationFocus™ CDx$_{BRCA\ LOH,}$ the median PFS of patients treated with rucaparib, who were determined to be *BRCA1/2*-positive, was 16.6 months as compared with 5.4 months for patients treated with placebo (HR: 0.23; 95% CI: 0.16, 0.34) (Table 5) [85, 86].

SOLO1 was a phase III, randomized, double-blind, placebo-controlled, multicenter trial that compared the efficacy of olaparib with placebo in patients with advanced ovarian,

**Table 5. Clinical utility of F1CDx for *BRCA1/2* alteration companion diagnostic claims.**

| Therapy | Clinical endpoint | F1CDx results | Full analysis set results |
|---|---|---|---|
| Olaparib | PFS[a], months | N = 206 | N = 260 |
| | HR[b] (95% CI) | Not reached | Not reached |
| | | 0.28 (0.20, 0.38) | 0.30 (0.23, 0.41) |
| Rucaparib | PFS[c], months | N = 124 | NR |
| | HR[b] (95% CI) | 16.6 | NR |
| | | 0.23 (0.16, 0.34) | |

CI = confidence interval; F1CDx = FoundationOne®CDx; HR = hazard ratio; NR = not reported; PFS = progression-free survival; RECIST = Response Evaluation Criteria in Solid Tumors.

[a] Investigator-assessed median PFS evaluated according to RECIST v1.1.

[a] HR for both F1CDx and full analysis set (olaparib only) compares olaparib or rucaparib to placebo for risk of disease progression or death.

[c] Investigator-assessed median PFS.

fallopian tube, or primary peritoneal cancer harboring a *BRCA* mutation (documented mutation in *BRCA1* or *BRCA2*) following first-line platinum-based chemotherapy [83]. A total of 391 patients were randomized (2:1) to receive olaparib tablets 300 mg orally twice daily (n = 260) or placebo (n = 131). The clinical validation of F1CDx was based on a clinical bridging retrospective analysis of 368 patients included in SOLO1 whose tumor samples were available for analysis, of which 313 patients were determined to have a documented mutation in *BRCA1/2* by F1CDx. The median PFS in both the F1CDx-positive patients and the full analysis set from the clinical trial were not reached at the time of analysis for patients treated with olaparib, as compared with 11.9 months and 13.8 months for placebo-treated patients, respectively [87]. The percent of progression events was similar between the full analysis set and the F1CDx determined set (39% in each group). When comparing olaparib to placebo for progression or death in this patient population, the HR was similar for those with *BRCA* mutation status as determined in the full analysis set (HR: 0.3; 95% CI; 0.23, 0.41) and those determined by F1CDx (HR: 0.28; 95% CI: 0.20, 0.38) (Table 5) [87].

### Clinical validation: Prostate cancer

**Clinical validation for detection of homologous recombination repair (HRR) mutations in prostate cancer.**  The clinical validity of F1CDx for determination of HRR status in metastatic castration-resistant prostate cancer (mCRPC) was determined with a prospective analysis of clinical data from the PROfound trial. PROfound was a phase 3, randomized, open-label, multicenter trial that assessed the efficacy and safety of olaparib monotherapy in patients with mCRPC who have qualifying HRR gene mutations that were predicted to be deleterious or suspected deleterious (known or predicted to be detrimental/lead to loss of function) and who have failed prior treatment with a novel hormonal agent (NHA) (investigators choice of NHA with either enzalutamide 160 mg orally once daily or abiraterone acetate 1000 mg orally once daily plus prednisone 5 mg orally twice daily [prednisolone was permitted for use instead of prednisone, if necessary]) [88]. Eligible patients were those with HRR gene mutation-positive (HRRm) mCRPC who had progressed following prior treatment with an NHA [88]. All patients must have had a qualifying HRR mutation assessed via the Foundation Medicine HRR CTA to be randomized [88]. Qualifying HRR gene mutations were in *BRCA1*, *BRCA2*, or *ATM* for Cohort A, and *BARD1*, *BRIP1*, *CDK12*, *CHEK1*, *CHEK2*, *FANCL*, *PALB2*, *PPP2R2A*, *RAD51B*, *RAD51C*, *RAD51D*, or *RAD54L* for Cohort B [88].

For the clinical utility results within Cohort A, the ORR for the full analysis set was similar to that of the F1CDx-determined HRRm patients (33.3% vs 33.8%) for olaparib-treated patients. Additional clinical efficacy results from Cohort A are reported in Table S30 in S1 Appendix. Further, the median radiological PFS (rPFS) was similar within this cohort for the full analysis set (7.4 months; 95% CI: 6.24, 9.33) and those HRRm patients as determined by F1CDx (7.4 months; 95% CI: 6.87, 9.33) for patients treated with olaparib, as compared to 3.6 months for those treated with NHA in both groups. The HR for the comparison of radiological progression or death for olaparib to placebo was also similar for the full analysis set (HR: 0.34; 95% CI: 0.25, 0.47) and for HRRm patients as determined by F1CDx (HR: 0.33; 95% CI: 0.24, 0.46). There was a statistically significant improvement in rPFS as assessed by the BICR for olaparib-treated patients compared with investigators choice of NHA-treated patients in Cohort A+B, with a 51% reduction in the risk of radiological disease progression or death and a prolongation of median progression-free interval of 2.3 months with olaparib vs investigator's choice of NHA (HR: 0.49; 95% CI 0.38, 0.63; *P*<0.0001) [87]. Median rPFS was 5.8 months for the full analysis set and 6.2 months for the confirmed F1CDx subgroup vs 3.5 months for investigator's choice of NHA (Table 6) [87].

**Table 6. Clinical utility of F1CDx for HRR mutations companion diagnostic claims in the combined Cohort A and Cohort B.**

| Clinical endpoint | F1CDx results | | Full analysis set results | |
|---|---|---|---|---|
| | Olaparib (n = 248) | Investigator's choice of NHA (n = 128) | Olaparib (n = 256) | Investigator's choice of NHA (n = 131) |
| rPFS[a], months | 6.2 | 3.5 | 5.8 | 3.5 |
| (95% CI) | (5.52, 7.36) | (2.10, 3.65) | (5.52, 7.36) | (2.20, 3.65) |
| HR[b] (95% CI) | 0.49 (0.38, 0.63) | | 0.49 (0.38, 0.63) | |

BICR = blinded independent central review; CI = confidence interval; F1CDx = FoundationOne®CDx; HR = hazard ratio; HRR = homologous recombination repair; ORR = overall response rate; PCWG3 = Prostate Cancer Working Group 3; RECIST = Response Evaluation Criteria in Solid Tumors; rPFS = radiological progression-free survival.

[a] rPFS based on BICR using RECIST v1.1 and/or PCWG3, or death (by any cause in the absence of progression) regardless of whether the patient withdrew from randomized therapy or received another anticancer therapy prior to progression.

[b] HR for both F1CDx and full analysis set compares olaparib to investigator's choice of therapy (either enzalutamide 160 mg orally once daily or abiraterone acetate 1000 mg orally once daily with prednisone 5 mg orally twice daily [prednisolone was permitted for use instead of prednisone, if necessary]) for radiological disease progression or death.

## Analytical performance validation

The performance characteristics of F1CDx were established using DNA from a wide range of FFPE tumor tissue types (Table S31 in S1 Appendix). Each study also included a broad range of representative alteration types for each class of alterations (SUBs, INDELs, CNAs, and rearrangements) in various genomic contexts across a broad selection of genes, as well as analysis of genomic signatures including MSI and TMB. The analytical validity of F1CDx was demonstrated across multiple analyses reporting the LoB, LoD, concordance with orthogonal method, and precision of the assay.

**Limit of blank.** The LoB describes the highest measurement result that is likely to be observed for a blank sample with a stated probability, α. Per industry standard, an α (type I error rate, false positive rate) of 0.05 was selected, corresponding to a percentage of false-positive <5%. The LoB of zero was confirmed using the mutation calls from mutation-negative FFPE samples (19 distinct samples with four replicates per sample). Seventy-five (75) samples (one sample failed to meet the Library Construction QC process specification) were used for the assessment of LoB. It was confirmed that each replicate of LoB sample was negative for the targeted variants defined in the LoD analysis, thus confirming the LoB of zero. A similar LoB study was conducted for each supplemental companion diagnostic claim; in each case, an LoB of zero was confirmed.

An additional study was conducted in a total of 218 evaluable test replicates for TMB-H status at a cut-off of ≥10 mut/Mb. The LoB was confirmed as zero. A supplementary statistical analysis was performed to assess the LoB for MSI-high (MSI-H). A total of 135 test replicates were evaluated confirming the LoB of zero for MSI-H.

**Limit of detection.** The LoD describes the precision of the assay at the lowest quantitative level at which an analyte (genomic variant) can be consistently detected. According to industry standard, "consistently detected" was defined as the level at which a 95% detection rate is observed. As defined by CLSI EP17A2E, the LoD is a "measured quantity value, obtained by a given measurement procedure, for which the probability of falsely claiming the absence of a measurand in a material is β, given a probability α of falsely claiming its presence" [91].

The LoD was determined for all F1CDx companion diagnostic claims. LoD was determined based on either allele frequency (as summarized in Table 7) or computational tumor purity (as summarized in Table 8). Allele frequency was the measurand for all SUBs and INDELs, wherein the frequency of an allele may be quantified. In contrast, the LoD for CNAs was based

**Table 7. Sample validation for F1CDx companion diagnostic claims.**

| Alteration | LoD[a] allele fraction (%) (hit rate) | LoD[b] allele fraction (%) (probit) |
|---|---|---|
| *EGFR* L858R | 2.4% | <2.4% (all detected) |
| *EGFR* Exon 19 deletion | 5.1% | 3.4% |
| *EGFR* T790M | 2.5% | 1.8% |
| *KRAS* G12/G13 | 2.3% | <2.3% (all detected) |
| *BRAF* V600E/K | 2.0% | <2.0% (all detected) |
| *MET* exon 14 SNVs[c] | 2.93% | <2.9% (all detected) |
| *MET* exon 14 insertion and deletion[c] | 5.73% | 5.7% |
| *BRCA1/2*[d] Alteration in non-repetitive or homopolymer <4 bp | Not calculated | 5.9% |
| Deletion in 8 bp homopolymer | Not calculated | 15.3% |
| HRR gene base substitutions | 5.44%–6.33%[e] | Not calculated |
| HRR gene INDELs | 5.22%–12.74%[e] | Not calculated |

*BRAF* = v-Raf murine sarcoma viral oncogene homolog B; *BRCA* = breast cancer gene; *EGFR* = epidermal growth factor receptor; HRR = homologous recombination repair; *KRAS* = V-Ki-ras2 Kirsten rat sarcoma; LoD = limits of detection; *MET* = mesenchymal-epithelial transition; *PIK3CA* = phosphatidylinositol-4,5-bisphosphate 3-kinase catalytic subunit alpha.

[a] LoD calculations for the companion diagnostic variants were based on the hit rate approach, as there were less than three levels with hit rate between 10% and 90% for all companion diagnostic variants (not including *BRCA*1/2 variants). LoD from the hit rate approach is defined as the lowest level with 95% hit rate (worst case scenario).

[b] LoD calculations for the companion diagnostic variants based on the probit approach with 95% probability of detection.

[c] For each sample, five levels of MAF, with 10 replicates per level, were evaluated for a total of 50 replicates per sample.

[d] See Summary of Safety and Effectiveness Data for P160018.

[e] LoD defined as the lowest level with ≥95% hit rate.

on the tumor purity of a given sample. DNA derived from FFPE tumor samples was evaluated for each companion diagnostic alteration or biomarker. For each LoD evaluation, five to six titrated levels of MAF or computational tumor purity, with 13 to 20 replicates per level, were prepared and assessed by F1CDx.

Further, the LoD for each variant category assessed by F1CDx was evaluated to determine the sensitivity of tumor profiling. Several categories of alteration types were evaluated, as detailed in Table S32 in S1 Appendix, as well as Table S33 in S1 Appendix for SVs and INDELs, and Table S34 in S1 Appendix for CNAs and copy number rearrangements. As with the companion diagnostic claims, DNA derived from FFPE tumor samples was selected for each of the variant categories. For each sample, five to six levels of MAF or computational tumor purity, with 13 to 20 replicates per level, were evaluated for a total of 78 to 94 replicates per sample.

The tumor profiling LoD by variant type as determined by the hit rate method is reported in Table S32 in S1 Appendix LoD for SVs, including SUBs and INDELs, is based on MAF. LoD for structural variants (fusions, amplifications, homozygous deletions, rearrangements) and complex biomarkers is based on computational tumor purity (described in more detail in the S2 Appendix). For platform-wide LoD, the INDELs (other than homopolymer repeat context) are grouped together as they had similar LoD characteristics. The INDELs ranged from 42bp insertions to 276bp deletions. INDELs at homopolymer repeat context had higher LoD, with a dependency on the length of the repeat context. Tumor profiling LoDs for alterations

**Table 8. Sample validation for F1CDx companion diagnostic claims.**

| Alteration | Tumor purity (%) (hit rate)[a] | Tumor purity (%) (probit)[b] |
|---|---|---|
| *ALK* fusion | 2.6%[c] | 1.8% |
| *ERBB2* amplification | 25.3%[d] | 19.7% |
| *BRCA2* homozygous deletion (HD) | 8.8%[e] | Not calculated |
| LOH[f] | 35% | 30% |
| *FGFR2* fusions | 5.31%[g] | 5.38% |
| HRR gene rearrangements[h] | 20.1%[g] | 26.0% |
| HRR gene homozygous deletions[h] | 23.9%[g] | 23.5% |
| TMB $\geq$ 10 mut/Mb[h] | 28.16%[g] | Not calculated |
| *NTRK1* fusions[i,j] | 12.1% | 12.6% |
| *NTRK2* fusions[i,k] | 11.5% | 11.6% |
| *NTRK3* fusions[i,l] | 6.1% | 3.7% |

*ALK* = anaplastic lymphoma kinase; *BRCA* = breast cancer gene; *ERBB2* = Erb-B2 receptor tyrosine kinase 2; *FGFR2* = fibroblast growth factor receptor 2; HRR = homologous recombination repair; Mb = megabase; mut, mutations; N/A = not applicable; *NTRK* = neurotrophic receptor tyrosine kinase; TMB = tumor mutational burden-high.

[a] Sensitivity calculations for the companion diagnostic variants were based on the hit rate approach, as there were less than three levels with hit rate between 10% and 90%. LoD from the hit rate approach is defined as the lowest level with 95% hit rate (worst scenario).

[b] Sensitivity calculations for the companion diagnostic variants based on the probit approach with 95% probability of detection.

[c] The number of chimeric reads for the sample evaluated is 16 at the indicated tumor fraction.

[d] The number of copy number amplifications for the sample evaluated is six at the indicated tumor fraction.

[e] The LoD calculation for the *BRCA2* HD was based on the hit rate approach, as there was a hit at every dilution level tested, making the probit regression not applicable.

[f] Please refer the F1CDx label for the Summary of Safety and Effectiveness Data for P160018/S001.

[g] Calculated using the 95% hit rate.

[h] For each sample, five levels of tumor purity, with 20 replicates per level except for the highest level at which 14 replicates were tested, were evaluated for a total of 94 replicates per sample.

[i] For each sample, a total of 94 tumor dilution replicates were assessed, including 20 replicates for each level of tumor purity, excluding the highest level, for which only 14 replicates were performed.

[j] The LoD study included two samples with CDx *NTRK1* fusion positive status: one *NTRK1-LMNA* fusion and one *NTRK1-TRP* fusion.

[k] The LoD study included two samples with CDx *NTRK2* fusion positive status: one *NTRK2-BCR* fusion and one *NTRK2-GARNL3* fusion.

[l] The LoD study included three samples with CDx *NTRK3* fusion positive status: three *NTRK3-ETV6* fusions.

based on allele frequency are further detailed in Fig 2A and Table S33 in S1 Appendix, and similarly, for alterations based on computational tumor purity in Fig 2B and Table S34 in S1 Appendix.

Detailed evaluation of SUBs demonstrated that the driver status (e.g., known, likely, and unknown), as assigned by the F1CDx analysis pipeline, has an important impact upon the calculated LoD, particularly for SUBs. Given the specific reporting rules in the analysis pipeline, the LoD for known driver SUBs is lower than the LoDs for likely and unknown SUBs, as depicted in Fig 3.

The LoD for TMB-H calling was evaluated using the hit rate method via two approaches: somatic TMB component variants by MAF; and tumor purity using cut-off at $\geq$10 mut/Mb. The resulting LoD (by MAF) for the TMB component variants within the interquartile range (IQR) ranged from 7.33% to 9.19% for SUBs, 7.08% to 11.74% for INDELs, and 7.32% to

A.

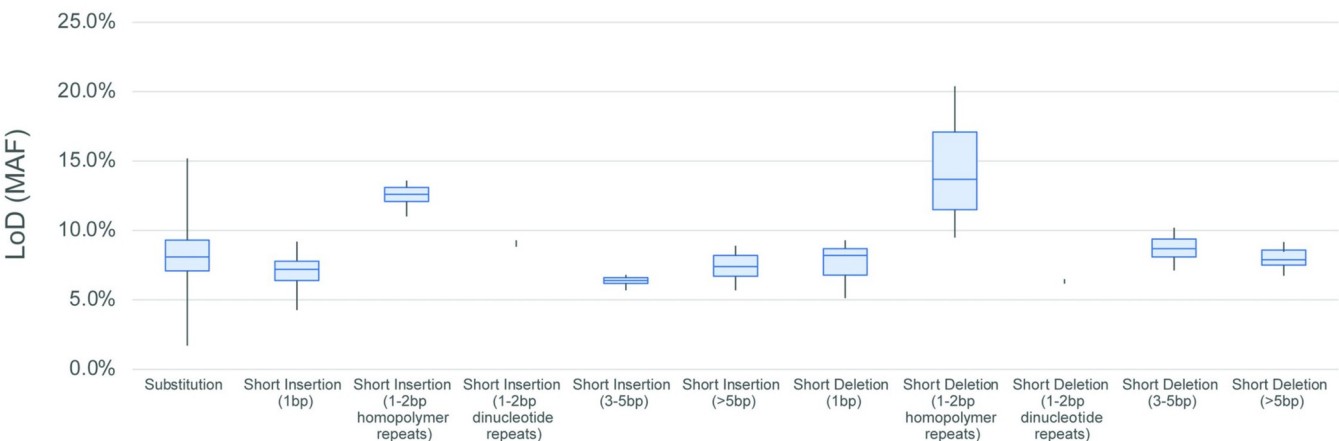

B.

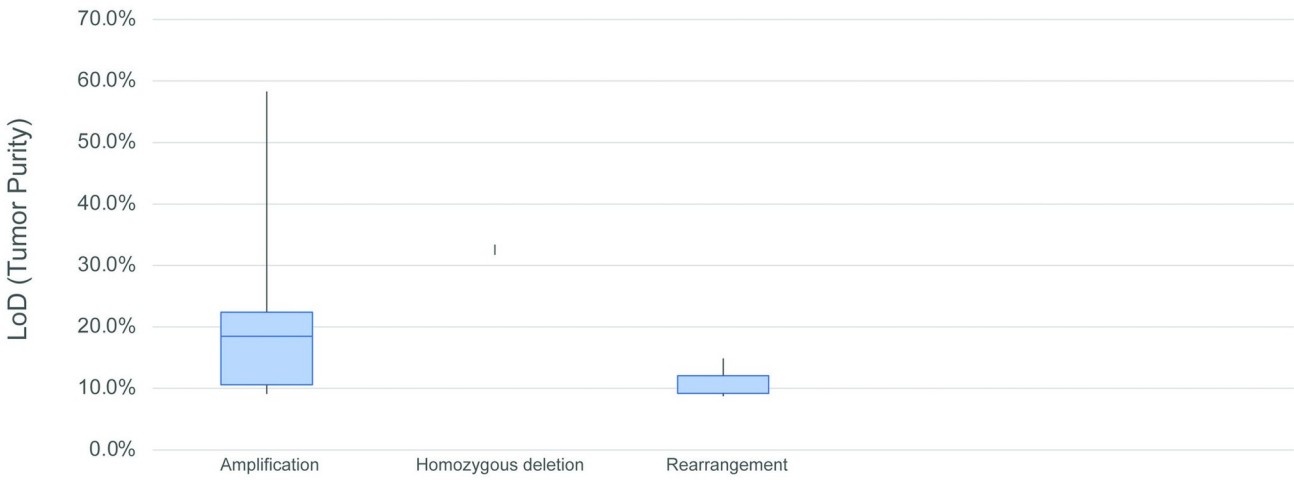

**Fig 2.** LoD ranges for (A) SVs and (B) amplifications and rearrangements [a] LoD calculations for the platform variants were based on the hit rate approach for variants with less than three levels with hit rate between 10% and 90% and probit approach for variants with at least three levels with hit rate between 10% and 90%. LoD from the hit rate approach is defined as the lowest level with 95% hit rate (worst-case scenario). [b] Data includes an alteration in the *TERT* promoter, 124C>T (LoD of 7.9%). *TERT* is the only promoter region interrogated and is highly enriched for repetitive context of poly-Gs, not present in coding regions. [c] Alterations classified as "known" are defined as those that are listed in COSMIC. [d] Alterations classified as "other" include truncating events in tumor suppressor genes (splice, frameshift, and nonsense) as well as variants that appear in hot-spot locations but do not have a specific COSMIC association or are considered VUS due to lack of reported evidence and conclusive change in function. [e] Sensitivity calculations for the platform variants were based on the hit rate approach for variants with less than three levels with hit rate between 10% and 90% and probit approach for variants with at least three levels with hit rate between 10% and 90%. LoD from the hit rate approach is defined as the lowest level with 95% hit rate (worst-case scenario). [f] Max represents VUS alteration at calling threshold. bp = base pair; CN = copy number; Mb = megabase; MSI = microsatellite instability; mut = mutation; TMB = tumor mutation burden; VUS = variants of unknown significance.

9.24% overall. The LoD for TMB-H calling was determined empirically by the hit rate method (defined as the lowest level with 95% hit rate) as 28.16% computational tumor purity. The LoD results using ≥10 mut/Mb cut-off is reported in Table S35 in S1 Appendix.

The LoD of MSI-H calling was evaluated using the hit rate method from five samples. Across the samples used for analysis, the highest observed tumor purity, with ≥95% of

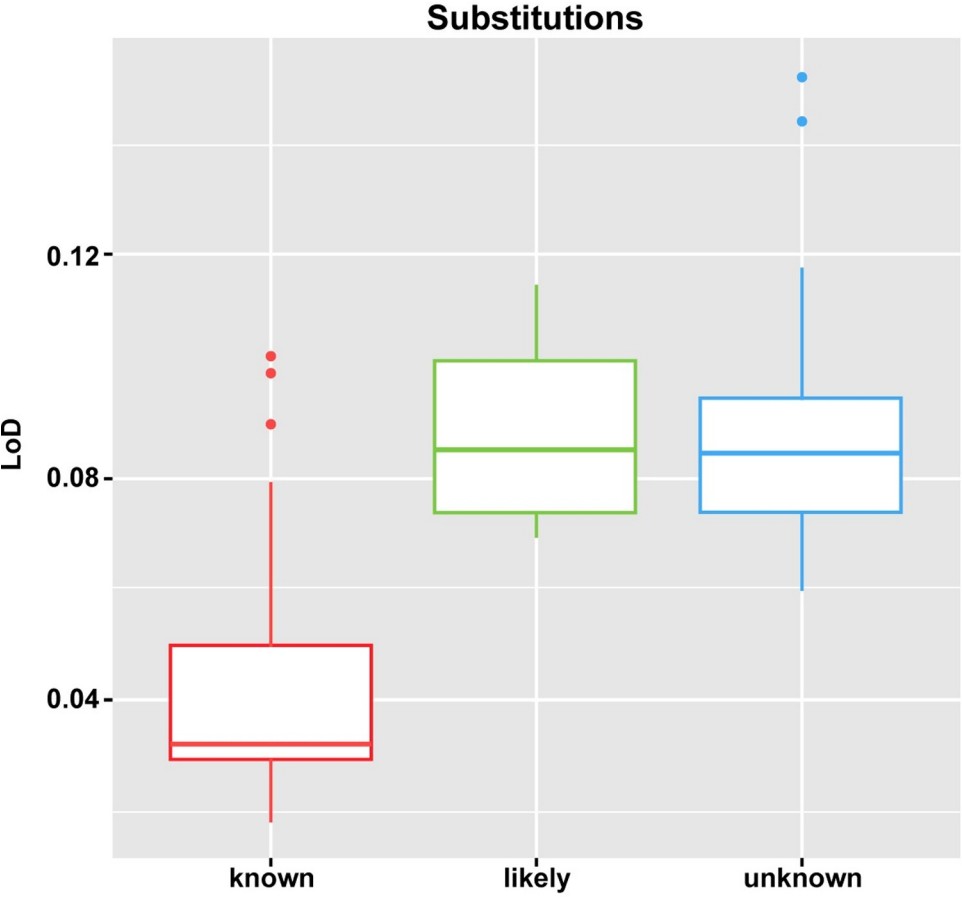

**Fig 3. LoDs per driver status (known, likely and unknown) for SUBs using the hit rate approach.** LoD = limit of detection.

replicates scored as MSI-H, was 15.67% tumor purity. The LoD for MSI-H is reported in Table S36 in S1 Appendix.

The LoD of gLOH calling in ovarian cancer samples was evaluated in four samples. The results validated a gLOH LoD (with 95% probability of detection) of 30% tumor purity, meeting the established acceptance criteria.

**Precision: Reproducibility and repeatability.** A total of 163 samples had alterations representative of companion diagnostic-associated alterations as well as exemplar alterations in a variety of genomic contexts. Each sample also included additional alterations that were included in the assessment. Across all samples, the pre-sequencing process failure rate was 1.5%, and the no-call rate was 0.18% for MSI-H, 6.38% for TMB-H (all samples), and 0.22% for TMB-H (samples with ≥10 mut/Mb). Within the assessment of repeatability and reproducibility for companion diagnostic variants, all variants from all samples demonstrated 100% agreement.

The reproducibility of calling hundreds of thousands of non-companion diagnostic platform alterations was evaluated across all sample replicates within the Precision study. The platform-level repeatability and reproducibility showed high overall agreement across alteration variants, and a high sample-level of reproducibility, as summarized in Tables 9 and 10 below. The platform-level study included a total of 443 SUBS, 188 INDELs, 55 CNAs, 13 copy number loss, and 18 rearrangements in the variant set across the samples. Additionally, the positive

**Table 9. Reproducibility of platform-wide variant detection.**

| Variant type | Unique variants | Concordant data points | Reproducibility (95% CI) |
|---|---|---|---|
| CNAs | 68 | 67,300 of 67,524 | 99.67% (99.62%, 99.71%) |
| Rearrangements | 18 | 17,851 of 17,874 | 99.87% (99.81%, 99.92%) |
| SUBs | 443 | 439,649 of 439,899 | 99.94% (99.94%, 99.95%) |
| INDELs | 188 | 186,319 of 186,684 | 99.80% (99.78%, 99.82%) |
| All Variants | 717 | 711,119 of 711,981 | 99.88% (99.87%, 99.89%) |

CI = confidence interval; CNA = copy number amplification; INDEL = insertion/deletion; SUB = substitution.

and negative call rates for the platform variants are reported in Table S37 in S1 Appendix. To supplement the previous study and further evaluate the precision for calling MSI-H and TMB-H in DNA derived from pan-tumor FFPE specimens, a supplementary analysis was

**Table 10. Reproducibility of companion diagnostic alterations.**

| Gene or biomarker | Tumor type | Number of unique samples | Alteration | Reproducibility (95% two-sided CIs[a]) |
|---|---|---|---|---|
| *EGFR* | NSCLC | 3 | Exon 19 deletion | 100.00% (96.50%, 100.00%) |
| | | 2 | Exon 21 L858R | 100.00% (94.73%, 100.00%) |
| | | 2 | Exon 20 T790M | 100.00% (94.93%, 100.00%) |
| *KRAS* | CRC | 3 | Codons 12/13 substitution | 100.00% (96.57%, 100.00%) |
| *ALK* | NSCLC | 3 | Fusion | 100.00% (96.57%, 100.00%) |
| *BRAF* | Melanoma | 2 | V600E/V600K | 100.00% (94.87%, 100.00%) |
| *ERBB2* | Breast cancer | 3 | Amplification | 100.00% (96.44%, 100.00%) |
| *BRCA1/2* | Ovarian cancer | 34 | SUBs, INDELs, rearrangements, homozygous deletions | 96.02% (94.49%, 97.14%) |
| *PIK3CA* | Breast cancer | 3[c] | E545K/ H1047L/C420R | 100.00% (95.58%, 100.00%) |
| *FGFR2* | Cholangiocarcinoma (CCA) | 5 | *FGFR2* Fusions and rearrangements[d] | 99.28% (96.01%, 99.87%) |
| *MET* | NSCLC | 8 | SNVs and INDELs that lead to exon 14 skipping | 98.95% (96.25%, 99.87%) |
| **HRR genes** | Prostate | 46 | SUBs, INDELs, rearrangements, homozygous deletions | 94.2% (92.9%, 95.4%) |
| **TMB** | Solid tumors | 46 | TMB ≥10 mutations per megabase | 99.72% (99.18%, 99.94%) |
| *NTRK1* | Solid tumors | 7[f] | Fusions | *NTRK1*: 100.0% (92.5%, 100.0%) |
| *NTRK2* | | | | *NTRK2*: 100.0% (85.2%, 100.0%) |
| *NTRK3* | | | | *NTRK3*: 99.2% (95.4%, 100.0%) |

CI = confidence interval; HRR = homologous recombination repair; LoD = limit of detection; PMA = premarket approval; sPMA = supplemental premarket approval; SUB = base substitution; TMB = tumor mutation burden.

[a] For *EGFR*, *KRAS*, *ALK*, *BRAF*, *ERBB2*, *BRCA1/2*, *PIK3CA*, and *FGFR2*, Wilson score method was used for 95% two-sided CIs calculation; for *MET*, HRR genes, TMB, and *NTRK1/2/3*, exact method was used for 95% two-sided CIs calculation.

[b] Six samples were evaluated, in which two samples had low reproducibility for calling *BRCA1/2* partial loss. This is expected as samples with small 1–2 exon partial losses were more challenging to detect.

[c] One sample is from the 46 samples originally included in the PMA precision study. One sample was analyzed in a subsequent precision study for the sPMA. An additional sample was analyzed in the associated PMC precision study. A *PIK3CA* H1047R alteration was also assessed, but excluded here because the alteration was present at only 0.15x the LoD; the reproducibility of that alteration event was 52.78%.

[d] The precision study included *FGFR2-BICC1*, *FGFR2-CCDC6* fusion; *FGFR2-TFCP2* fusion, and an intron 17 rearrangement (no partner).

[e] For HRR gene rearrangement, 15 variants were included; five variants exhibited reproducibility <90% due to read counts below reported LoD, which resulted in low reproducibility across all rearrangements of HRR genes.

[f] The precision study included seven samples with companion diagnostic *NTRK1/2/3* fusion positive status: four *NTRK3-EVT6* fusions, one *NTRK1-TPM3* fusion, one *NTRK1-LMNA* fusion, and one *NTRK2-DSTYK* fusion.

performed using 46 pan-tumor FFPE specimens. Repeatability and reproducibility results were 99.54% (95% CI: 98.39%, 99.89%) and 99.72% (95% CI: 99.18%, 99.94%), respectively, for the TMB-H cut-off of 10 mut/Mb (Table 10).

For the assessment of MSI, 100% repeatability and reproducibility was observed, with a lower limit of 99.7% and upper limit of 100%. Repeatability and reproducibility of MSI calling was robust. For MSI-H (using a cut-off at 0.0124), overall assessments of repeatability and reproducibility met acceptance criteria (i.e., ≥90% concordance). Repeatability and reproducibility across 46 samples were 96.02% (95% CI: 93.99%, 97.38%) and 97.01% (95% CI: 95.81%, 97.87%), respectively.

**Analytical concordance.** The detection of alterations by the F1CDx assay was compared to results of an externally validated NGS assay (evNGS) which was considered an orthogonal method for purposes of comparison. Concordance does not consider which method gives the "correct" result but only compares whether the methods used agree, and results can therefore be affected by technological and operational differences between the tests compared. The comparison between short alterations, including base SUBs and short INDELs, detected by F1CDx and the orthogonal method included 188 samples from 46 different tumors. The comparison of CNAs (including amplifications and homozygous deletions) and rearrangements detected between F1CDx and the orthogonal method included 168 samples. The samples analyzed for the orthogonal concordance for the companion diagnostic claims are further described in the Supplementary Appendix (S2 Appendix). A summary of PPA, NPA, and corresponding 95% two-sided exact CI is provided in Table 11.

The analysis for concordance of TMB-H (≥10 mut/Mb) detection was performed using a laboratory validated WES assay. This analysis evaluated 218 samples, of which 89 were not pre-screened by F1CDx (Set A) and 129 were pre-screened by F1CDx (Set B). Concordance results between F1CDx and WES for TMB-H calling are summarized in Table 12. The overall PPA and NPA were calculated based on a weighted average of the results (Set A and Set B) in the TMB-H concordance analysis. Overall PPA was 87.28% (95% CI: 64.42, 96.17), and overall NPA was 91.56% (95% CI: 85.66, 95.64).

Additionally, to support the use of retrospective data generated using FoundationOne®, a predecessor to F1CDx, a concordance study was conducted with F1CDx. This study evaluated a test set of 165 specimens. A total of 2,325 variants, including 2,026 SVs, 266 CNAs and 33 rearrangements, were included in the study. The study results are summarized in Table 13.

## 4. Discussion

In clinical application, CGP is designed to reduce the complexity of biomarker testing, enabling precision medicine to improve outcomes for patients with cancer. This improved outcome is predicated on test results being reliable, accurate, and validated to the highest standard available. The analyses presented here demonstrate the extensive analytical and clinical validation undertaken for F1CDx prior to initial FDA approval, and continuously over time to ensure reliability and relevance within an ever-changing clinical landscape.

Genomically-matched, biomarker-based precision therapies have been demonstrated to deliver significantly higher ORR across 13,203 patients in phase 1 and 32,149 patients in phase 2 trials, and in ORR, PFS, and overall survival (OS) in 38,104 patients in a meta-analysis of 112 registrational trials [92–94]. To assess CGP in informing the use of biomarker-based therapies, the clinical utility evidence base of Foundation Medicine CGP testing across tumor types has included assessment of actionability of test results and impact of test results on clinical decision-making (Table S38 in S1 Appendix) [4, 41, 93–101]. The addition of companion diagnostic claims is expected to expand upon this actionability and decision impact, and should be

**Table 11. Concordance of F1CDx and an externally validated NGS assay for platform-wide variants and CDx biomarkers.**

| Variant Type | F1CDx+/evNGS+ | F1CDx-/evNGS+ | F1CDx+/evNGS- | F1CDx-/evNGS- | PPA (95% CI)[a] | NPA (95% CI)[a] |
|---|---|---|---|---|---|---|
| All SVs | 1282 | 73 | 375 | 284218 | 94.6% (93.3%, 95.8%) | 99.9% (99.9%, 99.9%) |
| SUBs | 1111 | 39 | 334 | 242540 | 96.6% (95.4%, 97.6%) | 99.9% (99.8%, 99.9%) |
| INDELs | 171 | 34 | 41 | 41678 | 83.4% (77.6%, 88.2%) | 99.9% (99.9%, 99.9%) |
| All CNAs | 51 | 10 | 38 | 13845 | 83.6% (71.9%, 91.8%) | 99.7% (99.6%, 99.8%) |
| CNA: Amplification | 36 | 8 | 22 | 13878 | 81.8% (67.3%, 91.8%) | 99.8% (99.8%, 99.9%) |
| CNA: Homozygous deletions | 15 | 2 | 16 | 13911 | 88.2% (63.6%, 98.5%) | 99.9% (99.8%, 99.9%) |
| Rearrangements | 14 | 3 | 6 | 8713 | 82.4% (56.6%, 96.2%) | 99.9% (99.9%, 100%) |
| Total (CNAs and rearrangements) | 65 | 13 | 44 | 22,558 | 83.3% (73.2%, 90.8%) | 99.8% (99.7%, 99.9%) |
| *PIK3CA* substitutions in breast cancer | 53 | 0 | 0 | 48 | 100.00% (93.3%, 100.0%) | 100.00% (92.6%, 100.0%) |
| *FGFR2* fusions[b] | 25 | 2 | 1 | 130 | 87.08% (61.40%, 98.30%) | 99.59% (92.87%, 100.00%) |
| *MET* exon 14 SNVs and INDELs | 49 | 0 | 1 | 118 | 100.0% (92.8%, 100.0%) | 99.2% (95.4%, 100.0%) |
| HRR gene substitutions | 35 | 1 | 1 | 8243 | 97.22% (85.47%, 99.93%) | 99.99% (99.93%, 100.00%) |
| HRR gene INDELs | 75 | 6 | 2 | 17627 | 92.59% (84.57%, 97.23%) | 99.99% (99.96%, 100.00%) |
| HRR gene rearrangements | 10 | 1 | 5 | 1824 | 90.91% (58.72%, 99.77%) | 99.73% (99.36%, 99.91%) |
| HRR gene CNAs | 20 | 1 | 3 | 1356 | 95.24% (76.18%, 99.88%) | 99.78% (99.36%, 99.95%) |
| *NTRK1*, *NTRK2*, *NTRK3* fusions | 78[c,d] | 0[c,d] | 18[c,d] | 492[c,d] | 90.00% (75.00%, 100%)[c,f] | 99.94% (99.92%, 99.97%)[c,f] |
| | 16[c,e] | 2[c,e] | 0[c,e] | 20[c,e] | | |
| | 64[g,h] | 10[g,h] | 4[g,h] | 510[g,h] | 54.08% (37.94%, 71.37%)[g,j] | 99.98% (99.96%, 100.00%)[g,j] |
| | 15[g,i] | 3[g,i] | 0[g,i] | 20[g,i] | | |

CI = confidence interval; CNA = copy number alteration; CTA = clinical trial assay; evNGS = externally validated next-generation sequencing; F1CDx = FoundationOne®CDx; F1 LDT = FoundationOne® Laboratory Developed Test; FGFR2 = fibroblast growth factor receptor 2; HRR = homologous recombination repair; INDEL = insertion/deletion; NGS = next-generation sequencing; NPA = negative percent agreement; NTRK = neurotrophic receptor tyrosine kinase; PIK3CA = phosphatidylinositol-4,5-bisphosphate 3-kinase catalytic subunit alpha; PPA = positive percent agreement; SV = short variants; SUB = substitution.

[a] The PPA and NPA were calculated without adjusting for the distribution of samples enrolled using the F1 LDT; therefore, these estimates may be biased upward.

[b] PPA and NPA were adjusted using a prevalence of 9.6% to account for sampling differential.

[c] Primary analysis: a sample was considered as positive if an *NTRK1/2/3* rearrangement was detected; otherwise, it was considered as negative.

[d] Subset 1: samples where F1CDx served as the selection assay. Adjusted PPA and NPA based on an estimated prevalence of 0.32% in the intended use population to account for sampling differences were 100.00% (95% CI: 95.31%, 100.00%) and 99.94% (95% CI: 99.91%, 99.96%), respectively, based on the primary analysis.

[e] Subset 2: clinical trial samples where local clinical trial assays (LCTAs) served as the selection assay. PPA and NPA were 88.89% (95% CI: 67.20%, 96.90%) and 100.00% (95% CI: 83.89%, 100%), respectively, based on the primary analysis.

[f] The weighted PPA and NPA based on the bootstrapping of the combined dataset 10,000 times are shown for the primary analysis.

[g] Secondary analysis: a sample was considered F1CDx positive only if it met the *NTRK1/2/3* biomarker rule; otherwise, it was considered as F1CDx negative.

[h] Subset 1: samples where F1CDx served as the selection assay. Adjusted PPA and NPA based on an estimated prevalence of 0.32% in the intended use population to account for sampling differences were 13.58% (95% CI: 8.66%, 25.25%) and 99.98% (95% CI: 99.96%, 100.00%), respectively, based on the secondary analysis.

[i] Subset 2: clinical trial samples where local CTAs served as the selection assay. PPA and NPA were 83.33% (95% CI: 60.78%, 94.16%) and 100.00% (95% CI: 83.89%, 100%), respectively, based on the secondary analysis.

[j] The weighted PPA and NPA based on the bootstrapping of the combined dataset 10,000 times are shown for the secondary analysis.

included in assessment of clinical utility of the CGP approach. A growing body of evidence demonstrates clinical improvements in patient outcomes including PFS, OS, durable clinical response, and time to treatment failure (TTF) in patients whose treatments were informed by Foundation Medicine CGP, including F1CDx, in more recent studies (Table S39 in S1 Appendix) [4, 6, 94, 97, 101–105].

Foundation Medicine has demonstrated direct clinical validity and clinical utility for F1CDx in a pan-tumor setting for TMB-H status and *NTRK* fusions, and for *MET* exon 14 skipping in NSCLC, *PIK3CA* mutations in breast cancers, *BRCA1/2* alterations in ovarian

**Table 12. Concordance of F1CDx and an externally validated WES for TMB-High with ≥10 mut/Mb as cut-off.**

| | F1CDx+ /evWES+ | F1CDx-/evWES+ | F1CDx+ /evWES- | F1CDx-/evWES- | PPA (95% CI) | NPA (95% CI) |
|---|---|---|---|---|---|---|
| TMB ≥10 mut/Mb Set A | 28 | 7 | 4 | 50 | 80.00% (62.50, 90.62) | 92.59% (82.62, 98.04) |
| TMB ≥10 mut/Mb Set B[a] | 23 | 1 | 17 | 88 | 92.31% (65.74, 100.0) | 90.84% (87.76, 93.99) |

CI = confidence interval; evWES = externally validated whole exome sequencing; F1CDx = FoundationOne®CDx; Mb = megabase; mut = mutation; NPA = negative percent agreement; PPA = positive percent agreement; TMB = tumor mutation burden.

[s] PPA and NPA were adjusted using the prevalence of TMB-high estimated at 19%.

cancer, HRRm in prostate cancer, and *FGFR* fusions and rearrangements in cholangiocarcinoma. F1CDx has also demonstrated clinical validity for identifying *EGFR* exon 19 deletions, *EGFR* L858R and T790M mutated, *ALK*-rearranged or *BRAF* V600E-mutated tumors in NSCLC, *ERBB2* (HER2) amplifications in breast cancers, absence of *KRAS* mutations in CRC, and *BRAF* V600 mutations in melanoma demonstrating equivalent clinical utility to other validated assay in pivotal therapeutic trials.

Sequencing to a median depth of 500X across exons with ≥99% of exons at ≥100X coverage enables lower LoD and is particularly important for accurately reporting complex structural variants and signatures [36]. The F1CDx baitset devotes ~20% of the sequencing content to common germline SNPs distributed throughout the genome, which supports determination of a genome-wide CNA profile of each tumor and specifies the allelic balance of every segment. CNA modeling in combination with tumor purity and aneuploidy analysis enables downstream algorithms such as clinical gLOH [61].

Immune checkpoint inhibitors (ICPI) have transformed cancer care in select patient populations where these therapies are associated with clinical benefit. F1CDx can be supplemented by the incorporation of programmed death-ligand 1 (PD-L1) expression testing by immunohistochemistry in tumor types, which is relevant as a biomarker for ICPI treatment. Additional potential predictive biomarkers of ICPI resistance or hyperprogression that are evaluable by CGP include *JAK*/*STAT* pathway, *KEAP*, *STK11*, and *MDM2/4* amplifications [44, 106]. MSI-H and TMB-H (≥10 mut/Mb) have gained tumor-agnostic FDA approval as companion diagnostics for ICPI treatment. The importance of TMB-H was recently called into question as a biomarker for ICPI response [107]. However, as outlined in a Letter to the Editor, the failure

**Table 13. Concordance of F1CDx and FoundationOne® for platform-wide variants.**

| | F1CDx+/F1 LDT+ | F1CDx-/F1 LDT+ | F1CDx+/F1 LDT- | F1CDx-/F1 LDT- | PPA | NPA |
|---|---|---|---|---|---|---|
| **All variants** | 2246 | 33 | 46 | 322890 | 98.6% | 99.99% |
| **All SVs** | 1984 | 19 | 23 | 299099 | 99.1% | 99.99% |
| **SUBs** | 1692 | 10 | 19 | 254854 | 99.4% | 99.99% |
| **INDELs** | 292 | 9 | 4 | 44245 | 97.0% | 99.99% |
| **All CNA** | 230 | 14 | 22 | 19204 | 94.3% | 99.9% |
| **Amplifications** | 157 | 10 | 12 | 14671 | 94.0% | 99.9% |
| **Losses** | 73 | 4 | 10 | 4533 | 94.8% | 99.8% |
| **Rearrangements** | 32 | 0 | 1 | 4587 | 100.0% | 99.98% |

CNA = copy number alterations; F1CDx = FoundationOne®CDx; F1 LDT = FoundationOne® Laboratory Developed Test; INDEL = insertion/deletion; NPA = negative percent agreement; PPA = positive percent agreement; SV = short variants; SUB = substitutions.

to remove driver mutations during the *in-silico* cutoff analysis led to an overestimation of the TMB-H cohort, thereby diluting the clinical effect [56]. This highlights the importance of clinical and analytical validation of any assay intended to support therapy recommendations. TMB-H as a predictive biomarker for ICPI is supported by broad evidence, including a prospective cohort from the phase 2 KEYNOTE-158 study [57, 73, 108, 109].

Foundation Medicine's broad-panel, CGP approach meets the clinical needs of providers and patients to receive guideline-based biomarker testing and improves the paradigm over sequential single-gene testing that often leads to tissue exhaustion and extended times to identify and initiate treatment, thus helping to keep pace with a rapidly evolving field of medicine. Foundation Medicine's approach to CGP has been demonstrated to deliver improved clinical decision-making, and the single, high-throughput platform approach across diverse tumor types has demonstrated robust analytical and clinical validity and utility data [6, 58, 94, 97, 101–105, 110, 111]. Highly informative and personalized F1CDx results are provided with an assurance of quality in accordance with FDA and CLIA regulations and within clinically acceptable turnaround times. In accordance with the product requirements, a 10- to 14-day calendar turnaround is achieved in ≥90% of the samples processed, as measured by specimen receipt to issue date of the report (10 days). In practice, the median turnaround time in 2021 was 10.9 calendar days, and 90.1% of tests demonstrate turnaround ≤12 calendar days, from specimen receipt to issue date of the report (since June 19, 2021). This aligns with the CAP/ International Association for the Study of Lung Cancer/Association for Molecular Pathology recommended turnaround time of 10 working days or two calendar weeks [112].

While F1CDx reports identified deleterious variants, the test report intentionally does not provide germline mutation calling, which requires genetic counseling in many countries and may risk delaying therapeutic decision-making [113]. However, testing DNA derived from a tumor may capture both germline and somatic variants predictive of outcomes to targeted therapies [114]. The F1CDx clinical report highlights potential germline alterations relevant to cancer that may warrant additional germline testing in the appropriate clinical context.

Foundation Medicine's clinical reporting services also highlight genomic alterations that may have prognostic or diagnostic significance, such as *NUTM1* fusions in NUT carcinomas, *TMPRSS2-ERG* in prostate cancers, and *BRAF-KIAA1549* in gliomas, as well as context for low/intermediate MSI results in potential cases of Lynch Syndrome with high TMB [115–118]. Each specimen is reviewed by a licensed pathologist for tumor purity, suitability, and consistency with the stated diagnosis when provided, another crucial step in ensuring quality. Other Foundation Medicine decision support includes provision of molecular tumor board (MTB) services [63, 119, 120], allowing clinicians to tap the wealth of knowledge that a team of expert genomicists, pathologists, and oncologists provides. In a study assessing the impact of the Foundation Medicine MTB program, of 54 cases presented, 81% had one or more potentially therapeutically relevant alterations, and genomically-matched therapy or clinical trial options were offered to 46% of patients based on the MTB discussion [63]. A well-curated gene list has also allowed for review of recent results to inform physicians and patients of newly approved therapies for which their patients may be eligible. For example, when sotorasib was approved for NSCLC patients with *KRAS* G12C mutations, physicians with patients whose Foundation Medicine reports included this mutation in the previous three months were notified of sotorasib's approved indications.

Therapeutic decision-making is an increasingly complex challenge for practicing oncologists. The validation data presented here are the backbone for the ultimate patient impact provided by CGP. In patients who do undergo non-CGP molecular testing methods, such as single-gene tests, hot-spot panels, or cancer-specific focused panels (which typically rely on PCR or FISH methodology), limitations include: incomplete information, as not all relevant

genes and/or types of alterations nor genomic signatures are assessed [121–128]; inefficiency, as these methods may require sequential testing in certain cancer types [64, 124, 125, 129, 130]; and risk of re-biopsy, as multiple tests exhaust precious tissue [9, 64, 124, 125]. For example, CGP identifies genomic alterations missed by other testing methods in 41% to 84% of previously tested patients [15, 41].

As the number of actionable targets and associated targeted therapies expands, the capacity of most treating physicians to keep up with the precision oncology literature and guidelines will be exceeded. Therefore, to help support these crucial clinical choices, a streamlined testing process that allows a single test to identify all actionable alterations to be delivered together with a customized and timely literature review has already proven valuable to identify patients likely to respond to targeted therapy, and those patients bearing resistance markers who are not likely to respond [131]. This approach has already proven simpler and more sensitive than traditional sequential testing in screening CRCs for Lynch syndrome [132]. A test by itself does not directly impact patient response and survival outcomes; however, when properly validated, test results can inform appropriate treatment. Decision insights are significantly simplified in cancers with well-characterized mutations where companion diagnostics exist: a positive test result in the absence of contraindications indicates targeted therapy. This robust association of clinical benefit when therapeutic choice is predicated on a test has been achieved through hard-won clinical utility evidence from randomized controlled trials (RCTs) [49, 58, 83].

F1CDx has been assessed for FDA/CMS parallel review using the ACCE (Analytical validity, Clinical validity, Clinical utility, and Ethical, legal and social implications of genetic testing) model. FDA approval of F1CDx signified an acceptable evidence base for analytical and clinical validity requirements for coverage [133, 134]. CMS then carefully evaluated clinical utility by examining clinical outcome measures including OS, PFS, PR, CR, ORR, stable disease, time to progression, and TTF in systematic evidence reviews and meta-analyses of clinical trial data, including data from RCTs, and from nonrandomized studies [134, 135]. The CMS review resulted in Medicare coverage criteria for FDA-approved NGS-based *in vitro* companion diagnostic assays, like F1CDx, through a National Coverage Determination (NCD) [136]. F1CDx is covered under the NCD when the patient has:

a. Either recurrent, relapsed, refractory, metastatic, or advanced stages III or IV cancer; and

b. Either has not been previously tested using F1CDx for the same primary diagnosis of cancer or repeat testing using F1CDx only when a new primary cancer diagnosis is made by the treating physician; and

c. Decided to seek further cancer treatment.

Patients may not undergo molecular testing for many reasons, including lack of physician knowledge about the benefits of results to help inform treatment decision-making, lack of access to or reimbursement for testing, or other factors specific to the clinical scenario for a given patient, which make molecular testing utilizing a tissue-based test impractical. In a retrospective analysis of genomic testing patterns in NSCLC patients seen at 15 US sites between 2013 to 2015, only 8% (63/814) of eligible patients received broad-panel genomic profiling [9]. In 2017, a nationally representative survey of US oncologists showed that fewer than 17% reported ordering four or more commercially available NGS tests to guide treatment decisions during the previous 12 months, even though most were affiliated with academic institutions. Despite over half of respondents having reported some training in genomic testing, over half also reported difficulty in interpreting the results "some" or "most" of the time [137]. Most US patients with cancer are treated in the community setting, and the majority of oncologists use

some form of an NGS-based tumor testing to guide treatment decisions [137, 138]. In a nationwide survey of US physicians, 40% confirmed they obtain pharmacogenetic testing information from package inserts [139]. However, in an internal market research survey conducted by Foundation Medicine, US oncologists indicated that while 75%–80% of their patients with advanced cancer had received some sort of molecular diagnostic testing, only 25%–30% had received broad-panel molecular profiling. These broad-panel molecular profiling tests are used most often in indications like NSCLC or prostate cancer, where the presence of multiple biomarkers with approved targeted therapies creates additional need for broad-panel molecular profiling [140].

The inclusion of companion diagnostic and biomarker information in the therapeutic drug labeling is an important source of information for treating physicians. Likewise, the continued increase in the number of FDA-approved targeted therapies and use of CGP in earlier lines of therapy will create additional inroads into CGP diagnostic clinical application. Among >191,000 F1CDx reports delivered in the US from 2018 to 2020, the largest increases in reporting frequency were observed for: cemiplimab, approved in 2018 for advanced cutaneous

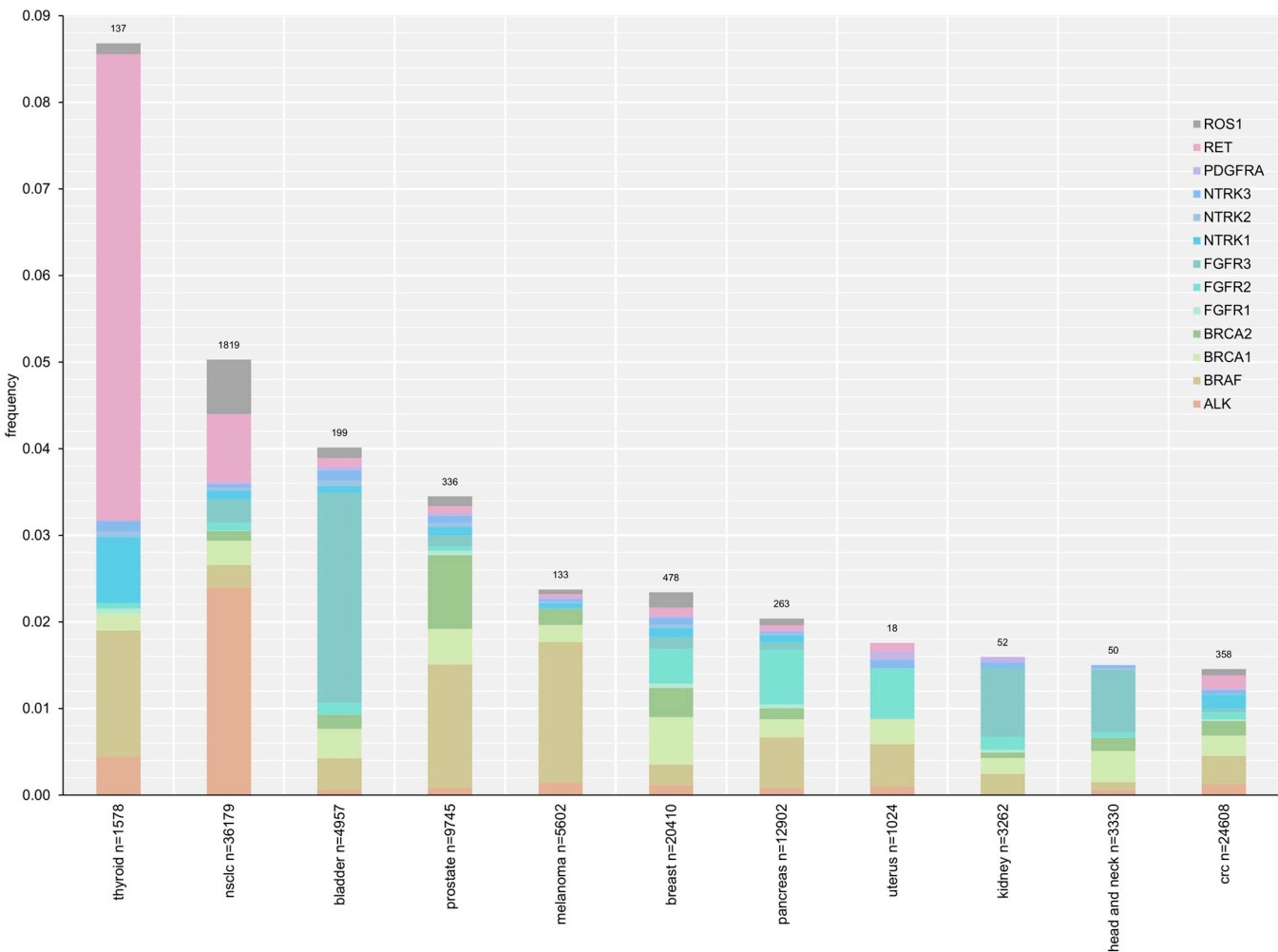

**Fig 4. Frequencies of targetable known/likely rearrangements in common tumor types detected by F1CDx between 2018 and 2020.** Source: from over 504,000 Foundation Medicine profiles consented for secondary research, 191,575 unique US patients tested with F1CDx from Jan. 14, 2018 through Mar. 31, 2021. CRC = colorectal cancer; NSCLC = non-small cell lung cancer.

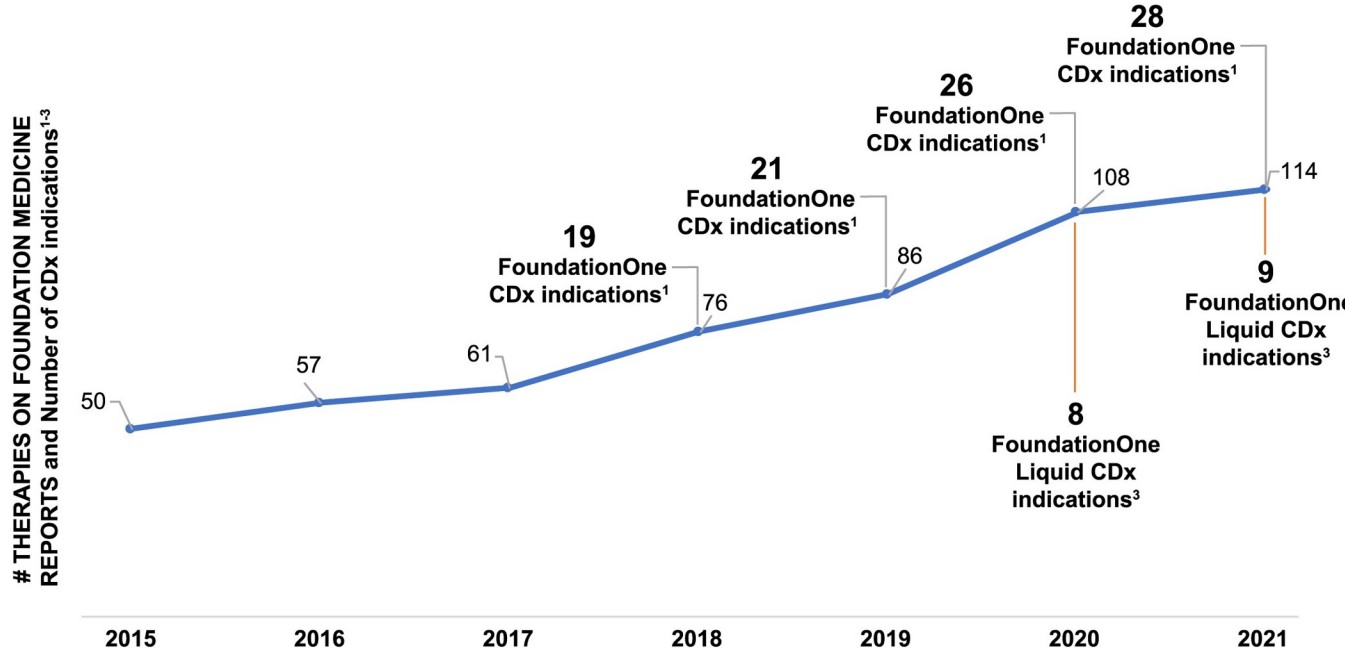

**Fig 5. Growth and evolution of precision medicine therapies and companion diagnostic biomarkers over time, as demonstrated by increasing number of approved therapies included on Foundation Medicine test results and increasing number of companion diagnostic indications and biomarkers.** [1] FoundationOne®CDx. Technical Information. Foundation Medicine, Inc; 2020. www.F1CDxLabel.com. [2] Data on File, Foundation Medicine, Inc., 2021. [3] FoundationOne®Liquid CDx. Technical Information. Foundation Medicine, Inc; 2020. www.F1LCDxLabel.com. Updated July 26, 2021. CDx = companion diagnostic.

squamous cell carcinoma [129] and in 2021 for advanced basal cell and NSCLC [130]; alpelisib, approved in 2019 for HR-positive, HER2-negative, *PIK3CA*-mutated advanced breast cancer [131]; selumetinib, approved in 2019 for pediatric neurofibromatosis type 1 plexiform neurofibromas [132]; talazoparib, approved in 2018 for germline *BRCA*-mutated HER2-negative advanced breast cancer [133]; fam-trastuzumab deruxtecan, approved in 2019 for third-line HER2-positive advanced breast cancer [134] and in 2021 for HER2-positive previously treated advanced gastric cancer [135]. Other recent approvals include capmatinib (approved in May 2020) [141] and tepotinib (approved in February 2021) [142] for metastatic NSCLC patients whose tumors exhibit splice site alterations in *MET* exon 14, a family of somatic alterations whose diversity and prevalence across diverse tumor types was first published by Foundation Medicine in 2015 [143].

Companion diagnostic biomarkers include alteration types in which the presence or the absence (e.g., wild-type) of the biomarker can inform treatment decisions. In addition, if other identified genomic alterations or genomic signatures may be associated with treatment resistance, the F1CDx report will include a note notifying of potential resistance, providing actionability to help rule out potentially ineffective treatment. Even negative results (i.e., the absence of actionable alterations) from CGP can provide benefit to patients and physicians, so long as testing methods are sensitive across a broad range of actionable alterations. For those without potentially matching therapy or clinical trial options, CGP may confirm chemotherapy as the best option and/or inform discussions about palliative care [144].

Rare or novel rearrangements and gene fusions are detected by F1CDx in many common tumor types. Fig 4 illustrates the frequency of known/likely rearrangements involving targetable genes in common tumors from >191,000 US F1CDx results since 2018. These results show that using only DNA sequencing, F1CDx reports targetable rearrangements in common

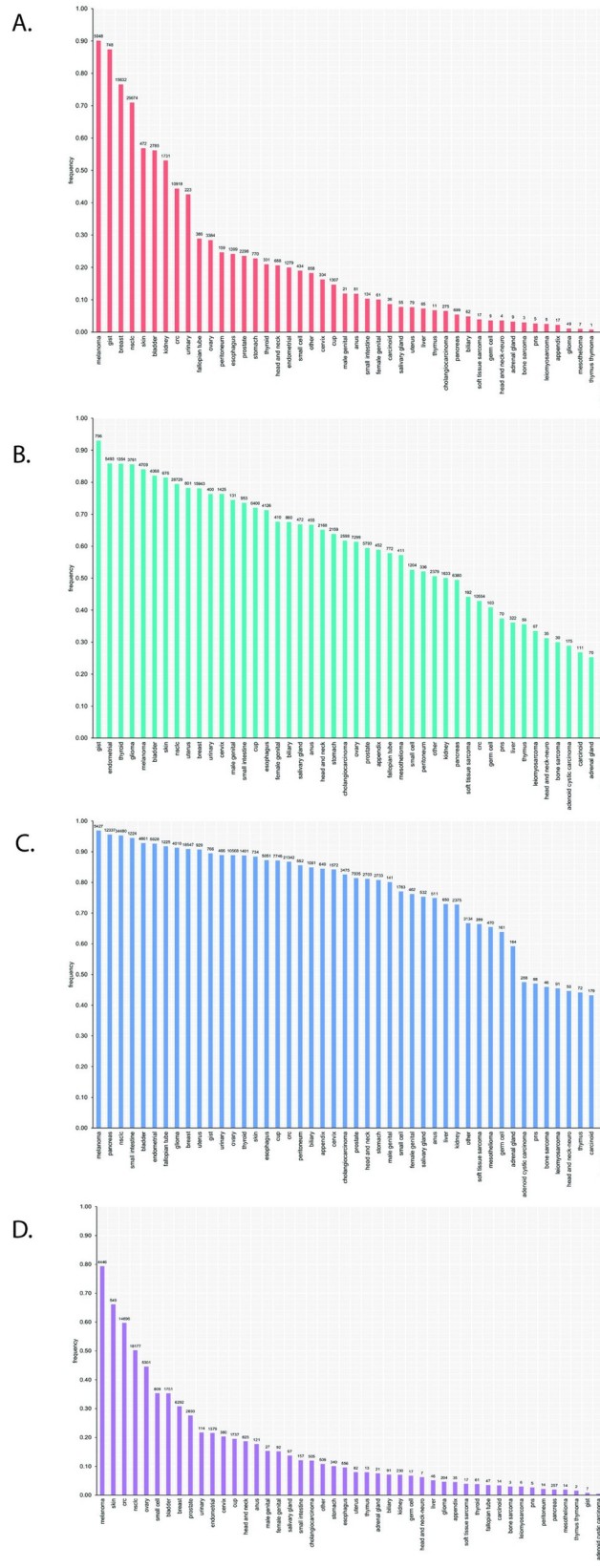

**Fig 6.** Frequency of F1CDx reports with potential therapeutic implications by disease group and definition of actionability: (A) therapy options available within the tumor type indicated; (B) therapy options available in tumor types other than the assigned indication; (C) disease groups with clinical trial options; and (D) disease groups with FDA approved companion/complementary diagnostics within the tumor type indicated. Source: 191,575 unique US patients tested with F1CDx and consented for secondary research from January 14, 2018 through March 31, 2021. All disease groups contain ≥100 specimens. Values indicate counts per disease group. CRC = colorectal cancer; CUP = cancer of unknown primary; GIST = gastrointestinal stromal tumor; NSCLC = non-small cell lung cancer; PNS = peripheral nervous system.

tumor types at rates consistent with those reported in the literature. Supplemental RNA sequencing is recommended for genomic profiling of hematologic, pediatric and rare malignancies and sarcomas, tumors in which fusions and gene rearrangements are more likely [145].

Clinical utility is challenging to assess in a rapidly evolving and complex field, and data from earlier studies (2014–2019) likely underestimate the potential therapeutic implications of F1CDx results today (Tables S38 and S39 in S1 Appendix). As evident in Fig 5, the number of targeted oncology therapy indications and approvals in oncology is growing rapidly [37], making the critical question in oncology care one of: *"Which therapy is appropriate for the patient?"*. Foundation Medicine CGP provides results that allow the patient and the physician to make informed treatment decisions based on evidence-based interventions that improve health outcomes and help keep pace with a rapidly evolving field. F1CDx reports delivered potential therapeutic implications in >191,000 reports from 2018 to 2020, overall and by disease group rank-ordered by therapy options in-tumor type, other tumor type, matching clinical trials, and companion/complementary diagnostics (Fig 6). In over 36,000 reports for NSCLC, ~50% included a companion diagnostic-associated therapy, ~70% included an in-tumor type therapy, ~80% included a therapy option in other tumor types, and nearly all included at least one relevant clinical trial. In over 20,000 breast cancer reports, ~30% included a companion diagnostic-associated therapy, ~76% included an in-tumor type therapy, ~76% included a therapy option in other tumor types, and ~90% include at least one relevant clinical trial. Refer to Table S40 in S1 Appendix for detailed data by tumor type.

Health economic outcomes data can provide additional evidence to demonstrate the value of CGP testing. Preliminary economic analyses have shown that the important clinical benefits of CGP may be accompanied by only a modest increase in cost, driven by longer duration of effective therapy for patients and meaningful prolongation of life which is the entire goal of cancer treatment [146–148]. As National Comprehensive Cancer Network (NCCN) states throughout its guidelines, "the best management of any patient with cancer is in a clinical trial" [28–34, 50–53]. Broad-panel genomic profiling has been associated with a 10% to 20% enrollment rate in clinical trials to date compared with a historical enrollment rate of ≤8% [41, 96, 98, 100, 102, 149], which may save payers up to $25,000 per patient through diversion of costs to study sponsors [18, 41, 96, 98, 100, 102].

The FDA has established a transparent process to validate tests in the high-risk setting of therapy selection, where poor performance translates to inferior patient outcomes. Ongoing studies will demonstrate even greater impact on decision-making and patient outcomes as a result of the FDA approval of F1CDx as a regulated companion diagnostic for an increasing number of targeted therapies. Regrettably, only a small minority of providers have chosen to invest in transparently demonstrating the rigor of their services, and aside from CMS, many payers are still able to choose cost over quality. As with pharmaceuticals, patients and providers need a reliable source of transparent performance data to be confident of the quality of the healthcare services they are receiving. Enabling precision medicine through guideline-driven adoption of only high-quality, validated tests will emerge as a more cost-effective and holistically beneficial treatment paradigm compared with traditional empirical treatment.

## Supporting information

**S1 Appendix. Tables S1–S40.**
(DOCX)

**S2 Appendix. Supplementary methods.**
(DOCX)

## Acknowledgments

The authors would like to acknowledge and thank all current and former Foundation Medicine employees who contributed to the development and launch of the F1CDx assay. With particular gratitude, we acknowledge the members of the product development, operations, and quality teams, in addition to our biopharma partners that contributed to analytical and clinical validation. Additional thanks are extended to Dawn Cardeiro for her assistance with compiling the decision impact data and Matt Carroll for analyzing the turnaround time for F1CDx processing. Further, the authors acknowledge Megan Pollack and her Xcenda colleagues for major contributions in compiling and editing this publication, as well as additional Foundation Medicine subject matter experts for their thorough review. This work was supported by Foundation Medicine, Inc.

## Author Contributions

**Conceptualization:** Coren A. Milbury, James Creeden, Wai-Ki Yip, David L. Smith, Varun Pattani, Kristi Maxwell, Bethany Sawchyn, Ole Gjoerup, Xiaobo Bai, Ninad Dewal, Dean C. Pavlick, Garrett M. Frampton, Christine Burns, Christine Vietz.

**Data curation:** Pei Ma.

**Formal analysis:** Wai-Ki Yip, Wei Meng, Pei Ma, Shannon T. Bailey, James Thornton.

**Investigation:** Joel Skoletsky, Alvin D. Concepcion, Yanhua Tang.

**Software:** Daniel Lieber, Jared White.

**Writing – original draft:** Coren A. Milbury, James Creeden, David L. Smith, Varun Pattani, Kristi Maxwell, Bethany Sawchyn, Ole Gjoerup, Daniel Lieber.

**Writing – review & editing:** Coren A. Milbury, James Creeden, Wai-Ki Yip, David L. Smith, Varun Pattani, Kristi Maxwell, Bethany Sawchyn, Ole Gjoerup, Wei Meng, Xiaobo Bai, Shannon T. Bailey, Dean C. Pavlick, Garrett M. Frampton, Jared White, Christine Burns, Christine Vietz.

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
