## [Decision Letter · Decision Letter 0]

7 Oct 2021

PONE-D-21-28288Clinical and analytical validation of FoundationOne®CDx, a comprehensive genomic profiling assay for solid tumorsPLOS ONE

Dear Dr. Milbury,

Thank you for submitting your manuscript to PLOS ONE. After careful consideration, we feel that it has merit but does not fully meet PLOS ONE’s publication criteria as it currently stands. Therefore, we invite you to submit a revised version of the manuscript that addresses the points raised during the review process.

ACADEMIC EDITOR: 

This is a MS with significant relevance. Please respond to the comments of 3 reviewers’ item-by-item to satisfy the reviewers' concerns. Please carefully edit the MS for English corrections and typos.

We look forward to receiving your revised manuscript.

Kind regards,

Nandini Dey, MS., Ph.D

Academic Editor

PLOS ONE

Journal Requirements:

"All authors received funding for this work from Foundation Medicine, Inc. This is the only funding source for this work."

"All authors are or were employees of Foundation Medicine, a wholly owned subsidiary of Roche, and have equity interest in Roche."

We note that you received funding from a commercial source: Roche

7. Thank you for submitting the above manuscript to PLOS ONE. During our internal evaluation of the manuscript, we found significant text overlap between your submission and the following previously published works, some of which you are an author.

https://www.accessdata.fda.gov/cdrh_docs/pdf17/P170019S017C.pdf

Please revise the manuscript to rephrase the duplicated text, cite your sources, and provide details as to how the current manuscript advances on previous work. Please note that further consideration is dependent on the submission of a manuscript that addresses these concerns about the overlap in text with published work.

8. We note that you are reporting an analysis of a microarray, next-generation sequencing, or deep sequencing data set. PLOS requires that authors comply with field-specific standards for preparation, recording, and deposition of data in repositories appropriate to their field. Please upload these data to a stable, public repository (such as ArrayExpress, Gene Expression Omnibus (GEO), DNA Data Bank of Japan (DDBJ), NCBI GenBank, NCBI Sequence Read Archive, or EMBL Nucleotide Sequence Database (ENA)). In your revised cover letter, please provide the relevant accession numbers that may be used to access these data. For a full list of recommended repositories, see http://journals.plos.org/plosone/s/data-availability#loc-omics or http://journals.plos.org/plosone/s/data-availability#loc-sequencing.

Additional Editor Comments:

This is a MS with significant relevance. Please respond to the comments of 3 reviewers’ item-by-item to satisfy the reviewers' concerns. Please carefully edit the MS for English corrections and typos.

Reviewers' comments:

Reviewer's Responses to Questions

**Comments to the Author**

1. Is the manuscript technically sound, and do the data support the conclusions?

Reviewer #1: Yes

Reviewer #2: Yes

Reviewer #3: Yes

2. Has the statistical analysis been performed appropriately and rigorously? 

Reviewer #1: N/A

Reviewer #2: Yes

Reviewer #3: Yes

3. Have the authors made all data underlying the findings in their manuscript fully available?

Reviewer #1: Yes

Reviewer #2: Yes

Reviewer #3: Yes

4. Is the manuscript presented in an intelligible fashion and written in standard English?

Reviewer #1: Yes

Reviewer #2: Yes

Reviewer #3: Yes

5. Review Comments to the Author

Reviewer #1: In this manuscript, Mibury et al. report Clinical and analytical validation of FundationOne. Since it is important to validate NGS sequencing data for clinical use, I agree to the authors’ point make of this manuscript. I think the manuscript is carefully written. Therefore, I basically do not have conceptual problem to this study. However, the manuscript is too long. I do not think people will read through the manuscript, except the reviewer like me, because peoples are usually interested in specific cancer types. In addition, the style of the manuscript is very unusual. I would recommend to revise this manuscript to “reader friendly style”.

Specific recommendation:

1: I would recommend to divide into several manuscripts.

2: Although the major part of the text in this manuscript is “Methods and Results”, this is very unusual. I somehow understand why the authors wrote with this style. However, as a reader, this makes hard to understand the study. I would recommend to write Methods and Results separately. In the result section, please just briefly describe their methods that are required to understand the results.

Reviewer #2: This is an important MS in the era of precision medicine. CGP is designed to provide accurate genetic alterations by virtue of reducing complexity of biomarker(s) testing and enabling precision medicine to improve outcome/management of cancer patients. May be it is important beyond the cancer patients.

This is an important MS but minor revision is needed before publication.

Minor comments:

1. Please discuss about variant of unknown significance (VUS), Tumor fraction (TF) and variations of allele frequencies (VAF)

2. Authors also provide a comparison between customized gene panel NGS vs WES vs WGS

3. Authors may also provide a simple methodology flow chart. It is good for the readers especially for new clinical researchers, residents and clinical fellow

4. Also need a table for abbreviations

Reviewer #3: Well written, carefully prepared validation Report of an important genetic test on its way from avant-garde to becoming gold standard. While there is no surprise and little novelty in this manuscript the data are important enough to be published and well discussed in a high quality journal.

Minor criticism:

To better appeal to the general reader the manuscript would profit from being shortened and becoming a bit more concise.

A list of abbreviations used should be included for easier readability.

6. PLOS authors have the option to publish the peer review history of their article (what does this mean?). If published, this will include your full peer review and any attached files.

Reviewer #1: No

Reviewer #2: No

Reviewer #3: No

---

## [Author Response · Author response to Decision Letter 0]

14 Dec 2021

https://journals.plos.org/plosone/s/file?id=wjVg/PLOSOne_formatting_sample_main_body.pdf [journals.plos.org] and 

https://journals.plos.org/plosone/s/file?id=ba62/PLOSOne_formatting_sample_title_authors_affiliations.pdf [journals.plos.org]

• We have reviewed this and confirm that we align with the style requirements.

"All authors received funding for this work from Foundation Medicine, Inc. This is the only funding source for this work."

• This research was funded by Foundation Medicine, Inc. The funder, Foundation Medicine, Inc. (a wholly owned subsidiary of Roche) provided support in the form of salaries for all authors (CAM, JC, WKY,DLS, VP, KM, BS, OG, WM, JS, ADC, YT, XB, ND, PM, STB, JT, DCP, GMF, DL, JW, CB, CV). The funders did not have any additional role in the study design, data collection and analysis, decision to publish, or preparation of the manuscript. The specific roles of these authors were provided in the cover letter at first submission. No grants supported this study.

"All authors are or were employees of Foundation Medicine, a wholly owned subsidiary of Roche, and have equity interest in Roche."

We note that you received funding from a commercial source: Roche

Within this Competing Interests Statement, please confirm that this does not alter your adherence to all PLOS ONE policies on sharing data and materials by including the following statement: "This does not alter our adherence to PLOS ONE policies on sharing data and materials.” (as detailed online in our guide for authors http://journals.plos.org/plosone/s/competing-interests). [journals.plos.org] If there are restrictions on sharing of data and/or materials, please state these. Please note that we cannot proceed with consideration of your article until this information has been declared. 

• The authors have the following interests. At the time of this research, all authors (CAM, JC, WKY,DLS, VP, KM, BS, OG, WM, JS, ADC, YT, XB, ND, PM, STB, JT, DCP, GMF, DL, JW, CB, CV) were employed by Foundation Medicine, Inc. (a wholly owned subsidiary of Roche), the funder of this study. This does not alter the authors' adherence to all the PLOS ONE policies on sharing data and materials, as detailed online in the guide for authors.

4. We note that you have indicated that data from this study are available upon request. PLOS only allows data to be available upon request if there are legal or ethical restrictions on sharing data publicly. For more information on unacceptable data access restrictions, please see http://journals.plos.org/plosone/s/data-availability#loc-unacceptable-data-access-restrictions. [journals.plos.org]

b) If there are no restrictions, please upload the minimal anonymized data set necessary to replicate your study findings as either Supporting Information files or to a stable, public repository and provide us with the relevant URLs, DOIs, or accession numbers. For a list of acceptable repositories, please see http://journals.plos.org/plosone/s/data-availability#loc-recommended-repositories [journals.plos.org].

• As confirmed with by each of the 3 reviewers, we have made all data underlying the findings in their manuscript fully available. Clinical and analytical validation included human research participant data. Institutional Review Board (IRB) approval was obtained from the New England IRB prior to use of samples in the described validation studies. All line data were provided to the FDA and are documented within in the publicly available FDA device summary of safety and effectiveness data (SSED). Our revised Data Availability Statement is provided below. 

• All relevant data were provided as supplementary information with this revised manuscript. In addition, detailed data may be obtained by contacting the corresponding author or the Foundation Medicine Data Governance Council at data.governance.council@foundationmedicine.com.

5. Please include captions for your Supporting Information files at the end of your manuscript, and update any in-text citations to match accordingly. Please see our Supporting Information guidelines for more information: http://journals.plos.org/plosone/s/supporting-information. [journals.plos.org]

• We have updated this per request.

• We have checked the reference list per request. No changes have been made to the reference list.

7. Thank you for submitting the above manuscript to PLOS ONE. During our internal evaluation of the manuscript, we found significant text overlap between your submission and the following previously published works, some of which you are an author.

https://www.accessdata.fda.gov/cdrh_docs/pdf17/P170019S017C.pdf [accessdata.fda.gov]

Please revise the manuscript to rephrase the duplicated text, cite your sources, and provide details as to how the current manuscript advances on previous work. Please note that further consideration is dependent on the submission of a manuscript that addresses these concerns about the overlap in text with published work.

• On October 12, 2021, we inquired about the overlap in text between our submission and the following previously published works, specifically with regard to our device Summary of Safety and Effectiveness Data (SSED) (https://www.accessdata.fda.gov/cdrh_docs/pdf17/P170019S017C.pdf [accessdata.fda.gov]). The SSED for our device is required by the FDA. The SSED is not a peer-reviewed manuscript, but rather the public posting of our data to demonstrate safety and effectiveness. We explained this is a common occurrence, as all FDA-approved drugs and devices are required to have their data available publicly. As an example, our sister assay (the FoundationOne Liquid CDx assay) is quite similar, where the validation methods and results are published both in your peer-reviewed journal (https://10.1371/journal.pone.0237802 [doi.org] ), as well as posted online via the FDA SSED (https://www.accessdata.fda.gov/cdrh_docs/pdf20/P200016B.pdf [accessdata.fda.gov] ). On October 20, 2021, we received a response back from Alexis Miller (Publications Assistant, PLOS ONE), that after consulting with an Academic Research Associate, PLOS ONE notes that this overlap is acceptable and that we may disregard this note.

8. We note that you are reporting an analysis of a microarray, next-generation sequencing, or deep sequencing data set. PLOS requires that authors comply with field-specific standards for preparation, recording, and deposition of data in repositories appropriate to their field. Please upload these data to a stable, public repository (such as ArrayExpress, Gene Expression Omnibus (GEO), DNA Data Bank of Japan (DDBJ), NCBI GenBank, NCBI Sequence Read Archive, or EMBL Nucleotide Sequence Database (ENA)). In your revised cover letter, please provide the relevant accession numbers that may be used to access these data. For a full list of recommended repositories, see http://journals.plos.org/plosone/s/data-availability#loc-omics [journals.plos.org] or http://journals.plos.org/plosone/s/data-availability#loc-sequencing [journals.plos.org].

• Clinical and analytical validation included human research participant data. Institutional Review Board (IRB) approval was obtained from the New England IRB prior to use of samples in the described validation studies. All line data were provided to the FDA and are documented within in the publicly available FDA device summary of safety and effectiveness data (SSED). Our revised Data Availability Statement is provided below. 

• All relevant data were provided as supplementary information with this revised manuscript. In addition, detailed data may be obtained by contacting the corresponding author or the Foundation Medicine Data Governance Council at data.governance.council@foundationmedicine.com.

• Patient-level data derived from the Foundation Medicine research dataset cannot be shared or deposited as they contain patient genomic information that, depending on the prevalence of the identified alterations, could be used to identify individuals.

Additional Editor Comments:

This is a MS with significant relevance. Please respond to the comments of 3 reviewers’ item-by-item to satisfy the reviewers' concerns. Please carefully edit the MS for English corrections and typos.

Reviewers' comments:

Reviewer's Responses to Questions

Comments to the Author

1. Is the manuscript technically sound, and do the data support the conclusions?

Reviewer #1: Yes

Reviewer #2: Yes

Reviewer #3: Yes

2. Has the statistical analysis been performed appropriately and rigorously? 

Reviewer #1: N/A

Reviewer #2: Yes

Reviewer #3: Yes

3. Have the authors made all data underlying the findings in their manuscript fully available?

The PLOS Data policy [plosone.org] requires authors to make all data underlying the findings described in their manuscript fully available without restriction, with rare exception (please refer to the Data Availability Statement in the manuscript PDF file). The data should be provided as part of the manuscript or its supporting information, or deposited to a public repository. For example, in addition to summary statistics, the data points behind means, medians and variance measures should be available. If there are restrictions on publicly sharing data—e.g. participant privacy or use of data from a third party—those must be specified.

Reviewer #1: Yes

Reviewer #2: Yes

Reviewer #3: Yes

4. Is the manuscript presented in an intelligible fashion and written in standard English?

Reviewer #1: Yes

Reviewer #2: Yes

Reviewer #3: Yes

5. Review Comments to the Author

Reviewer #1: In this manuscript, Mibury et al. report Clinical and analytical validation of FundationOne. Since it is important to validate NGS sequencing data for clinical use, I agree to the authors’ point make of this manuscript. I think the manuscript is carefully written. Therefore, I basically do not have conceptual problem to this study. However, the manuscript is too long. I do not think people will read through the manuscript, except the reviewer like me, because peoples are usually interested in specific cancer types. In addition, the style of the manuscript is very unusual. I would recommend to revise this manuscript to “reader friendly style”.

Specific recommendation:

1: I would recommend to divide into several manuscripts.

2: Although the major part of the text in this manuscript is “Methods and Results”, this is very unusual. I somehow understand why the authors wrote with this style. However, as a reader, this makes hard to understand the study. I would recommend to write Methods and Results separately. In the result section, please just briefly describe their methods that are required to understand the results.

• Reviewer #1 requested that the manuscript be shortened and made more concise. Extensive compression of the manuscript was performed. Clinical validation text content was condensed and presented in concise table format. These revisions shortened the main manuscript text by approximately 10 pages and 7 tables were removed and replaced by 2.

• Reviewer #1 recommended that the manuscript be divided into several manuscripts because they felt it was too long. We feel that dividing this manuscript would not align with our intent, which is to summarize all validation data that support our FDA-approved comprehensive tissue test in order to demonstrate the aggregated strength of comprehensive genomic profiling. Further, reviewer #1 recommended that the methods and results be separated, and additional method sections are added (thus, increasing the length). On October 12, 2021, we inquired about whether the current formatting was acceptable. On October 20, 2021, we received a response back from Alexis Miller (Publications Assistant, PLOS ONE), that the manuscript adheres to PLOS ONE guidelines of formatting and in its current format is eligible for publication. 

Reviewer #2: This is an important MS in the era of precision medicine. CGP is designed to provide accurate genetic alterations by virtue of reducing complexity of biomarker(s) testing and enabling precision medicine to improve outcome/management of cancer patients. May be it is important beyond the cancer patients.

This is an important MS but minor revision is needed before publication.

Minor comments:

1. Please discuss about variant of unknown significance (VUS), Tumor fraction (TF) and variations of allele frequencies (VAF)

2. Authors also provide a comparison between customized gene panel NGS vs WES vs WGS

3. Authors may also provide a simple methodology flow chart. It is good for the readers especially for new clinical researchers, residents and clinical fellow

4. Also need a table for abbreviations

• Reviewer #2 requested additional discussion on variants of unknown significance, tumor fraction, and variations of allele frequency. Additional text to summarize VUS has been added in the Bioinformatic Methods section. Tumor Fraction is not applicable to the solid tumor F1CDx assay (this is a characteristic that is unique to liquid biopsies). We have however, expanded discussion to clarify computational tumor purity; refer to the new text in the Bioinformatic Methods section. 

• Reviewer #2 requested additional discussion for how customized gene panel NGS differ from WES and WGS. Revisions and additional content were added within the Introduction section. 

• Reviewer #2 requested that a simple flow chart be provided. Figure 1 presents the simplified flow chart for the F1CDx methodology. 

• Reviewers #2 requested that a table of acronyms and definitions be included. This has been provided as Table S1.

Reviewer #3: Well written, carefully prepared validation Report of an important genetic test on its way from avant-garde to becoming gold standard. While there is no surprise and little novelty in this manuscript the data are important enough to be published and well discussed in a high quality journal.

Minor criticism:

To better appeal to the general reader the manuscript would profit from being shortened and becoming a bit more concise.

A list of abbreviations used should be included for easier readability.

• Reviewer #3 requested that a table of acronyms and definitions be included. This has been provided as Table S1.

• Reviewer #3 requested that the manuscript be shortened and made more concise. Extensive compression of the manuscript was performed. Clinical validation text content was condensed and presented in concise table format. These revisions shortened the main manuscript text by approximately 10 pages and 7 tables were removed and replaced by 2.

 6. PLOS authors have the option to publish the peer review history of their article (what does this mean? [journals.plos.org]). If published, this will include your full peer review and any attached files.

Do you want your identity to be public for this peer review? For information about this choice, including consent withdrawal, please see our Privacy Policy [plos.org].

Reviewer #1: No

Reviewer #2: No

Reviewer #3: No

---

## [Decision Letter · Decision Letter 1]

4 Feb 2022

Clinical and analytical validation of FoundationOne®CDx, a comprehensive genomic profiling assay for solid tumors

PONE-D-21-28288R1

Dear Dr. Creeden,

We’re pleased to inform you that your manuscript has been judged scientifically suitable for publication and will be formally accepted for publication once it meets all outstanding technical requirements.

Kind regards,

George Vousden

Staff Editor

PLOS ONE

Additional Editor Comments (optional):

Reviewers' comments:

Reviewer's Responses to Questions

**Comments to the Author**

1. If the authors have adequately addressed your comments raised in a previous round of review and you feel that this manuscript is now acceptable for publication, you may indicate that here to bypass the “Comments to the Author” section, enter your conflict of interest statement in the “Confidential to Editor” section, and submit your "Accept" recommendation.

Reviewer #1: All comments have been addressed

Reviewer #2: All comments have been addressed

Reviewer #3: All comments have been addressed

2. Is the manuscript technically sound, and do the data support the conclusions?

Reviewer #1: Yes

Reviewer #2: Yes

Reviewer #3: Yes

3. Has the statistical analysis been performed appropriately and rigorously? 

Reviewer #1: N/A

Reviewer #2: I Don't Know

Reviewer #3: N/A

4. Have the authors made all data underlying the findings in their manuscript fully available?

Reviewer #1: Yes

Reviewer #2: Yes

Reviewer #3: Yes

5. Is the manuscript presented in an intelligible fashion and written in standard English?

Reviewer #1: Yes

Reviewer #2: Yes

Reviewer #3: Yes

6. Review Comments to the Author

Reviewer #1: (No Response)

Reviewer #2: Revised MS is much better and informative than the original one. After revision the MS is much improved and good for Physician scientists and researchers. Also after adding the flow chart in the methodology, it is much easy to follow.

Reviewer #3: In the revised manuscript the authors have made a good job in making it shorter and more concise without loosing valuable information. The paper will give the field and even clinicians the possibility to gain additional and developing insight into tools they are starting to use in their daily practice.

7. PLOS authors have the option to publish the peer review history of their article (what does this mean?). If published, this will include your full peer review and any attached files.

Reviewer #1: No

Reviewer #2: No

Reviewer #3: **Yes: **Prof. Dr. Michael W. Krainer, Medical University Vienna

---

## [Editor Report · Acceptance letter]

8 Mar 2022

PONE-D-21-28288R1 

Clinical and analytical validation of FoundationOne®CDx, a comprehensive genomic profiling assay for solid tumors 

Dear Dr. Creeden:

I'm pleased to inform you that your manuscript has been deemed suitable for publication in PLOS ONE. Congratulations! Your manuscript is now with our production department. 

Kind regards, 

on behalf of

Dr. George Vousden 

Staff Editor

PLOS ONE